# Geometry-aware Score Distillation via 3D Consistent Noising and Gradients

## Abstract

Score distillation sampling (SDS), the methodology in which the score from pre-trained 2D diffusion models is distilled into 3D representation, has recently brought significant advancements in text-to-3D generation task. However, this approach is still confronted with critical geometric inconsistency problems such as the Janus problem. Starting from our observation that such inconsistency problems are induced by multiview inconsistencies between 2D diffusion scores predicted from various viewpoints, we introduce **G**eometry-aware **S**core **D**istillation (**GSD**), a simple and general plug-and-play framework for incorporating 3D consistency and therefore geometry awareness into the SDS process. Our methodology is composed of three components: 3D consistent noising, designed to produce 3D consistent noise maps that follow the standard Gaussian distribution, geometry-based gradient warping for identifying correspondences between predicted gradients of different viewpoints, and gradient consistency loss to optimize the scene geometry toward producing more consistent gradients. We demonstrate that our plug-and-play technique applied on various baseline score distillation-based methods significantly improves performance, successfully addressing the geometric inconsistency problems with minimal computation cost.

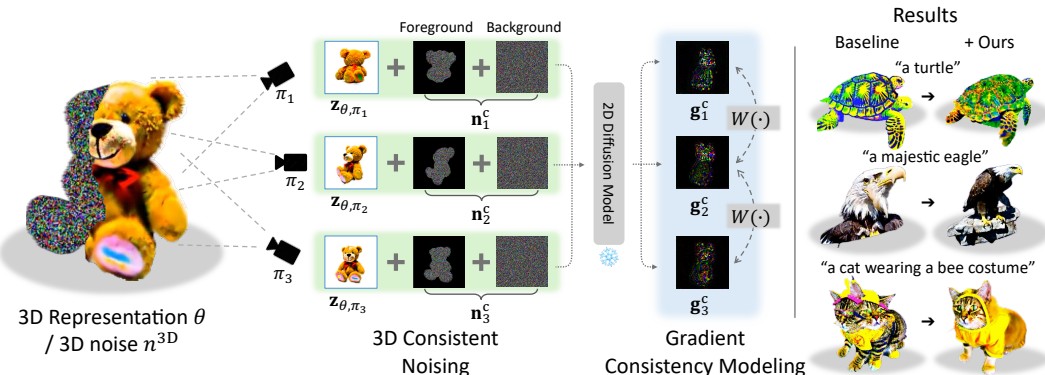

Figure 1: **Teaser.** Our framework incorporates 3D awareness into the score distillation sampling (SDS) process through a 3D consistent noising and gradient consistency modeling, which improves consistency of the 2D diffusion scores predicted from various viewpoints. As a general plug-and-play module that can be attached to any SDS-based text-to-3D generation baselines (Poole et al., 2023; Yi et al., 2023) with little computation cost, it brings about highly enhanced view consistency and fidelity to 3D generation results.

## 1 Introduction

Text-to-3D generation, which is the task of generating a 3D scene from a text prompt, has seen great advancements in recent years due to the advent of powerful generative models such as diffusion model (Ho et al., 2020; Song et al., 2020). As the main objective of this task is to generate a high-quality 3D model solely from user-given text, it enables even non-professional users to create 3D models easily with little to no handwork. Naturally, advancements in this task have opened up numerous possibilities in various domains such as VR/AR, computer-generated graphics, and gaming.

However, due to the limited size and quantity of 3D ground truth datasets compared to 2D images or videos, directly training a diffusion model on 3D representations is challenging. To address this, most methods (Jain et al., 2022; Lin et al., 2023; Chen et al., 2023a; Wang et al., 2023) use pretrained 2D diffusion models to optimize 3D representations (Mildenhall et al., 2020; Müller et al., 2022; Kerbl et al., 2023) through score distillation sampling (SDS) (Poole et al., 2023), where the 3D representation is refined using the 2D diffusion model's score from noised scene renderings at various viewpoints. However, since the 2D diffusion model lacks explicit knowledge of 3D domain, it often results in geometric inconsistencies like the Janus problem (Seo et al., 2024; Shi et al., 2023), where multi-faced geometries harm the global shape, making it unsuitable for real-world applications.

To understand and counter this issue, we analyze the SDS process from the perspective of multiview consistency, observing that such geometric inconsistency problem is correlated to the independence of each SDS process, which in turn causes the lack of multiview consistency between 2D scores predicted from different viewpoints. More specifically, we focus on the fact that under the naïve SDS setting (Poole et al., 2023), a single point in 3D receives vastly different optimization signals from various viewpoints, resulting artifacts and geometrically inconsistent geometric features such as Janus problem. Under this observation, encouraging the multiview consistency of 2D diffusion scores between nearby viewpoints would lead to reduction in such artifacts.

In this light, we propose a novel methodology, named **G**eometry-aware **S**core **D**istillation (**GSD**), which incorporates multiview correspondence awareness to the SDS process to facilitate multiview consistency of predicted gradients, as described in Fig. 1. Our method is a **plug-and-play** module that can be attached to existing SDS-based baselines for enhanced geometric consistency, with little computation cost and no need for additional networks or modules. Our method consists of three components. First, to encourage multiview consistency of predicted 2D scores across viewpoints, we introduce 3D consistent noising, combining point cloud representation with integral noising (Chang et al., 2024) to produce 3D geometry-aware 2D Gaussian noises in SDS process. Our 3D consistent noising imbues separate SDS denoising processes implicitly with 3D awareness. Secondly, we propose geometry-based gradient warping to warp the generated gradient of a viewpoint to other viewpoints, allowing for the comparison of gradients between corresponding locations across various viewpoints. We finally leverage the warped gradients for our novel multiview gradient consistency loss, which helps to regularize and reduce inconsistent scene features such as the Janus problem.

Our experimental results and analysis show that the application of our methodology strongly benefits the optimization process across various SDS-based text-to-3D baselines (Yi et al., 2023; Tang et al., 2024; Poole et al., 2023). Our methodology enhances the geometric consistency and fidelity of the generated results, resulting 3D scenes competitive to state-of-the-art. Our ablation study demonstrates that our contributions are strongly interconnected, justifying the need for all our components to be used in conjunction with one another.

## 2 RELATED WORK

**Text-to-3D generation.** DreamFusion (Poole et al., 2023) and SJC (Wang et al., 2022) introduced an optimization technique called score distillation sampling (SDS), which leverages pretrained large-scale text-to-image diffusion models to generate 3D objects. Since its introduction, SDS has been widely adopted in various text-to-3D generation models. Magic3D (Lin et al., 2023) and Fantasia3D (Chen et al., 2023b) employ a coarse-to-fine strategy with SDS optimization, achieving high-fidelity results. ProlificDreamer (Wang et al., 2023) has significantly improved the quality of 3D objects generated from text-to-3D tasks. This progress is due to treating the model's 3D parameters as random variables instead of constants, as in SDS, and developing a gradient-based update rule using the Wasserstein gradient flow. More recently, models such as DreamGaussian (Tang et al., 2024), GSGEN (Chen et al., 2023b), LucidDreamer (Liang et al., 2023) and GaussianDreamer (Yi et al., 2023) incorporates 3D Gaussian Splatting representation into SDS-based text-to-3D generation.

**Geometric inconsistency problem within SDS.** In text-to-3D generation tasks, maintaining 3D geometric consistency is crucial, yet a geometric inconsistency problem called the Janus problem (Wang et al., 2023; Shi et al., 2023) commonly occurs. Various approaches have been attempted address this. Multi-view Diffusion models such as MVDream (Shi et al., 2023) and EfficientDreamer (Zhao et al., 2023) fine-tunes a pretrained Stable Diffusion (Rombach et al., 2022) model using a 3D dataset

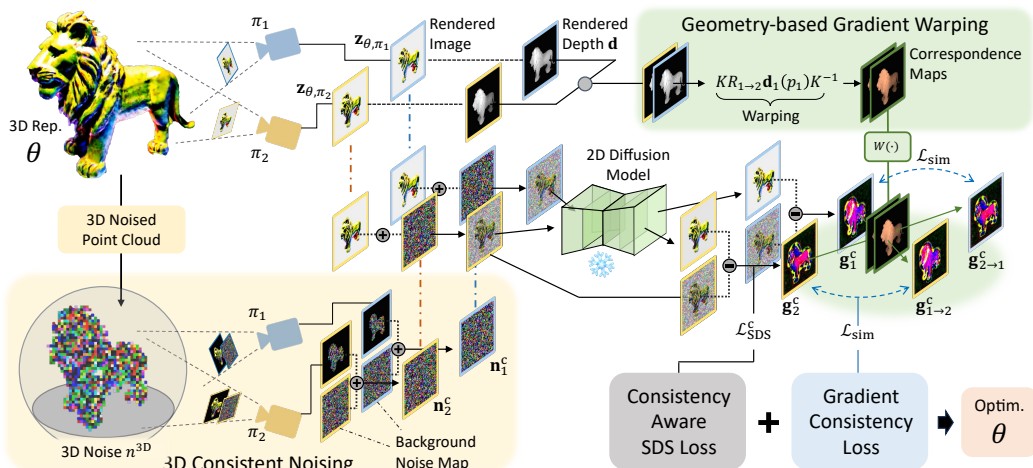

Figure 2: **Overall framework.** Our framework consists of three components for geometry-aware score distillation: 3D consistent noising, geometry-based gradient warping, and gradient consistency modeling. Through these components, our framework encourages multiview consistency between predicted 2D scores and enhances the quality of generated 3D scenes.

and enabled the model to generate orthogonal multi-view images with robust geometric consistency. 3DFuse (Seo et al., 2024) proposes a method that injects coarse 3D priors into a pretrained diffusion model. However, MVDream and EfficientDreamer rely on a large-scale 3D dataset Objaverse (Deitke et al., 2023) during training, which is limited in terms of asset quality, causes the model to generate clay-textured images similar to those in the Objaverse dataset. 3DFuse is also limited in another aspect, still exhibiting numerous 3D geometric inconsistencies depending on the coarse 3D priors.

## 3 Preliminaries

Diffusion models have demonstrated impressive capabilities in text-to-image generation (Nichol et al., 2021; Saharia et al., 2022; Ahn et al., 2024). Building on this achievement, DreamFusion (Poole et al., 2023) introduces the score distillation sampling (SDS), which generates plausible 3D objects by leveraging pretrained text-to-image diffusion models to optimize 3D representation such as NeRF (Mildenhall et al., 2020) parameterized by $\boldsymbol{\theta}$. SJC (Wang et al., 2022) formulates this SDS based on the assumption that a **3D probability density** of $\boldsymbol{\theta}$ given prompt $y$, denoted by $p_{\sigma_t}(\boldsymbol{\theta}; y)$, is proportional to the expected probability densities of multiview 2D rendered images $\mathbf{z}_{\boldsymbol{\theta},\pi}$ over the camera poses $\pi$ sampled from the distribution of the camera viewpoints $\Pi$, denoted by $p_{\sigma_t}(\mathbf{z}_{\boldsymbol{\theta},\pi}; y)$, where $\sigma_t$ denotes a noise level at time step $t$. This can be expressed as $\mathbb{E}_t\left[p_{\sigma_t}(\boldsymbol{\theta}; y)\right] \propto \mathbb{E}_{\pi \sim \Pi, t}\left[p_{\sigma_t}(\mathbf{z}_{\boldsymbol{\theta},\pi}; y)\right]$. The score is the gradient of the log probability density of data, so the following equation is derived using Jensen's inequality, with $\log \tilde{p}_{\sigma_t}(\boldsymbol{\theta}; y)$ as the lower-bound of $\log p_{\sigma_t}(\boldsymbol{\theta}; y)$:

$$\nabla_{\boldsymbol{\theta}} \mathcal{L}_{\text{SDS}} := \mathbb{E}_t\left[\underbrace{\nabla_{\boldsymbol{\theta}} \log \tilde{p}_{\sigma_t}(\boldsymbol{\theta}; y)}_{\text{3D score}}\right] = \mathbb{E}_{\pi \sim \Pi, t}\left[\underbrace{\nabla_{\mathbf{z}_{\boldsymbol{\theta},\pi}} \log p_{\sigma_t}(\mathbf{z}_{\boldsymbol{\theta},\pi}; y)}_{\text{2D score}} \cdot \frac{\partial \mathbf{z}_{\boldsymbol{\theta},\pi}}{\partial \boldsymbol{\theta}}\right], \qquad (1)$$

where the **2D score**, or the gradient of $\log p_{\sigma_t}(\mathbf{z}_{\boldsymbol{\theta},\pi}; y)$, is obtained using pretrained 2D diffusion models, e.g., pretrained Stable Diffusion (Rombach et al., 2022).

However, instead of directly using the rendered image $\mathbf{z}_{\boldsymbol{\theta},\pi}$, perturb-and-average scoring (PAAS) is required due to out-of-distribution problems, in which the 2D noise $\mathbf{n} \sim \mathcal{N}(0, \mathbf{I})$ is added to $\mathbf{z}_{\boldsymbol{\theta},\pi}$. Specifically, it defines the denoiser $\mathcal{D}(\cdot)$ such that $\mathcal{D}(\mathbf{z}_{\boldsymbol{\theta},\pi} + \sigma_t \mathbf{n}; \sigma_t, y) = (\mathbf{z}_{\boldsymbol{\theta},\pi} + \sigma_t \mathbf{n}) - \sigma_t \boldsymbol{\epsilon}_\phi(\mathbf{z}_{\boldsymbol{\theta},\pi} + \sigma_t \mathbf{n}, y, t)$ with the rendered image from $\boldsymbol{\theta}$ at camera pose $\pi$, aggregated with noise $\mathbf{n}$ scaled by noise level $\sigma_t$. The residual noise $\boldsymbol{\epsilon}_\phi(\cdot)$ is predicted from a frozen 2D diffusion model (Rombach et al., 2022) parameterized by $\phi$. It then defines a gradient map $\mathbf{g}_{\boldsymbol{\theta},\pi}$ representing the **2D score** as follows:

$$\mathbf{g}_{\boldsymbol{\theta},\pi} = \frac{\mathcal{D}(\mathbf{z}_{\boldsymbol{\theta},\pi} + \sigma_t \mathbf{n}; \sigma_t, y) - (\mathbf{z}_{\boldsymbol{\theta},\pi} + \sigma_t \mathbf{n})}{\sigma_t^2}, \qquad (2)$$

and when we compute expectation over these predicted gradients w.r.t random noise $\mathbf{n}$, it gives us the score, or the update direction, for the non-noisy rendered image $\mathbf{z}_{\boldsymbol{\theta},\pi}$ itself:

$$\nabla_{\mathbf{z}_{\boldsymbol{\theta},\pi}} \log p_{\sqrt{2}\sigma_t}(\mathbf{z}_{\boldsymbol{\theta},\pi}) \approx \mathbb{E}_{\mathbf{n}\sim\mathcal{N}(0,\mathbf{I}),t}\left[\mathbf{g}_{\boldsymbol{\theta},\pi}\right]$$

$$= \mathbb{E}_{\mathbf{n}\sim\mathcal{N}(0,\mathbf{I}),t}\left[\frac{\mathcal{D}(\mathbf{z}_{\boldsymbol{\theta},\pi} + \sigma_t\mathbf{n}; \sigma_t, y) - \mathbf{z}_{\boldsymbol{\theta},\pi}}{\sigma_t^2}\right] - \underbrace{\mathbb{E}_{\mathbf{n}\sim\mathcal{N}(0,\mathbf{I}),t}\left[\frac{\mathbf{n}}{\sigma_t}\right]}_{=0}, \quad (3)$$

where $\log p_{\sqrt{2}\sigma_t}(\cdot)$ appears because the diffusion model predicts the Gaussian noise of already noised $\mathbf{z}_{\boldsymbol{\theta},\pi}$, and as $\mathbb{E}_{\mathbf{n}\sim\mathcal{N}(0,\mathbf{I})}\left[\mathcal{N}(\mathbf{z}_{\boldsymbol{\theta},\pi} + \sigma_t\mathbf{n}; \mu, \sigma_t^2\mathbf{I})\right] = \mathcal{N}(\mathbf{z}_{\boldsymbol{\theta},\pi}; \mu, 2\sigma_t^2\mathbf{I})$, the variance becomes $2\sigma_t^2$ in regards to $\mathbf{z}_{\boldsymbol{\theta},\pi}$ and thus resulting a logarithm with the base of $\sqrt{2}\sigma_t$ (Wang et al., 2022).

Relating back to Eq. 1, obtaining a 3D score for optimizing $\boldsymbol{\theta}$ requires computing the expectation over multiple camera viewpoints $\pi$. Assuming a rendered image $\mathbf{z}_{\boldsymbol{\theta},\pi}$ at the viewpoint $\pi$ that is noised with noise $\mathbf{n}$, the final equation for score distillation is expressed as follows:

$$\nabla_{\boldsymbol{\theta}}\mathcal{L}_{\text{SDS}} \approx \mathbb{E}_{\pi\sim\Pi, \mathbf{n}\sim\mathcal{N}(0,\mathbf{I}),t}\left[\frac{\mathcal{D}(\mathbf{z}_{\boldsymbol{\theta},\pi} + \sigma_t\mathbf{n}; \sigma_t, y) - \mathbf{z}_{\boldsymbol{\theta},\pi}}{\sigma_t^2} \cdot \frac{\partial\mathbf{z}_{\boldsymbol{\theta},\pi}}{\partial\boldsymbol{\theta}}\right]. \quad (4)$$

## 4 METHODOLOGY

### 4.1 MOTIVATION AND OVERVIEW

In the standard SDS process (Poole et al., 2023; Wang et al., 2022; 2023), the 2D noise $\mathbf{n}$ is sampled independently per viewpoint. This raises questions about cases where two sampled viewpoints are close together, resulting in the rendered images $\mathbf{z}_{\boldsymbol{\theta},\pi}$ overlapping regions. Under the SDS setting, the different renderings of the overlappings would result in largely unrelated 2D scores for supervision, as the noises $\mathbf{n}$ are sampled independently. Put simply, it *lacks multiview consistency*. Our work starts from this observation that such a lack of multiview consistency induces geometric inconsistency problems such as the Janus problem. We seek to counter this problem by incorporating geometric awareness into the SDS process (Poole et al., 2023; Wang et al., 2022; 2023).

Assume a mapping function $\mathcal{W}(\cdot)$ that holds the 3D correspondences between viewpoints. Given the explicit 3D geometry represented by $\boldsymbol{\theta}$, we can obtain $\mathcal{W}(\cdot)$ by identifying which locations in 2D renderings correspond to the same point in 3D space, establishing geometry-based correspondence across different viewpoints. This $\mathcal{W}(\cdot)$ can then be used to map an image from one viewpoint to another in a geometrically consistent way – a process known as warping. Intuitively, applying $\mathcal{W}_{j\rightarrow i}(\cdot)$ to the noise $\mathbf{n}_j \sim \mathcal{N}(0,\mathbf{I})$ at viewpoint $\pi_j$ and mapping it to nearby viewpoint $\pi_i$ would result in multiview-consistent noise $\mathcal{W}_{j\rightarrow i}(\mathbf{n}_j)$ for $\mathbf{z}_{\boldsymbol{\theta},\pi_i}$. We observed that this approach ultimately yields more similar and aligned 2D scores between the two viewpoints. The gradient map $\mathbf{g}_{\boldsymbol{\theta},\pi_i}^{\text{w}}$ predicted from viewpoint $\pi_i$ is defined as:

$$\mathbf{g}_{\boldsymbol{\theta},\pi_i}^{\text{w}} = \sum_{\pi_j\in\Pi_{i,j}} \frac{\mathcal{D}(\mathbf{z}_{\boldsymbol{\theta},\pi_i} + \sigma_t\mathcal{W}_{j\rightarrow i}(\mathbf{n}_j); \sigma_t, y) - (\mathbf{z}_{\boldsymbol{\theta},\pi_i} + \sigma_t\mathcal{W}_{j\rightarrow i}(\mathbf{n}_j))}{\sigma_t^2}, \quad (5)$$

where $\Pi_{i,j}$ denotes the set of camera poses near an anchor pose $\pi_i$. The equation for multiview consistent SDS loss is then defined as follows:

$$\nabla_{\boldsymbol{\theta}}\mathcal{L}_{\text{SDS}}^{\text{w}} \approx \mathbb{E}_{\pi_i\sim\Pi, \mathbf{n}_j\sim\mathcal{N}(0,\mathbf{I}),t}\left[\frac{\mathcal{D}(\mathbf{z}_{\boldsymbol{\theta},\pi_i} + \sigma_t\mathcal{W}_{j\rightarrow i}(\mathbf{n}_j); \sigma_t, y) - \mathbf{z}_{\boldsymbol{\theta},\pi_i}}{\sigma_t^2} \cdot \frac{\partial\mathbf{z}_{\boldsymbol{\theta},\pi_i}}{\partial\boldsymbol{\theta}}\right], \quad (6)$$

assuming the warped noise map $\mathcal{W}_{j\rightarrow i}(\mathbf{n}_j)$ retains the properties of the standard normal distribution (*ref.* Section 4.2). Note that $\mathbf{z}_{\boldsymbol{\theta},\pi_i}$ can also be approximated as $\mathbf{z}_{\boldsymbol{\theta},\pi_i} \approx \mathcal{W}_{j\rightarrow i}(\mathbf{z}_{\boldsymbol{\theta},\pi_j})$. This means that the nearer the viewpoints are and $\mathcal{W}_{j\rightarrow i}$ approaches identity mapping, the estimated scores of nearby viewpoints in Eqn. 5 and Eq. 6 also increase in similarity and consistency. Based on Chang et al. (2024), which shows that incorporating correspondence relationships between the noises significantly enhances video generation quality, we hypothesize that maintaining consistency between the noises and gradients across multiple viewpoints would similarly benefit the optimization process, leading to more robust and coherent geometry.

In this paper, we propose **GSD**, a general framework for facilitating the multiview consistency of 2D scores predicted through SDS, improving the geometric consistency and fidelity of generated scenes, as shown in Fig. 2. In Section 4.2, we introduce **3D consistent noising**, which grounds each viewpoint's denoising process on the 3D geometry of the given scene. In Section 4.3, we conduct geometry-based gradient warping across different viewpoints. In Section 4.4, we describe our **correspondence-aware gradient consistency loss** exploiting the warped gradients, which effectively regularizes artifacts and inconsistencies by modeling multiview consistency of the 2D scores.

## 4.2 3D CONSISTENT NOISING

We propose a 3D consistent noise that incorporates 3D correspondence prior $\mathbf{n}^c$, enabling robust 3D scene generation. This promotes more consistent 2D scores across different viewpoints, as described in Fig. 3. A key factor in designing $\mathbf{n}^c$ is that the 2D noise produced by consistent noising should follow a standard normal distribution – namely, its mean and variance being that of $\mathcal{N}(0, \mathbf{I})$, and the noise should be independently and identically distributed (i.i.d.).

This makes the naïve solution of warping a 2D noise to another viewpoint using $\mathcal{W}(\cdot)$ unsuitable, as the

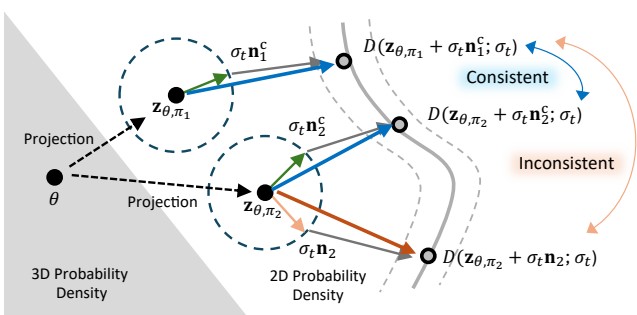

Figure 3: **PAAS-based illustration of our consistent noising.** Introduction of 3D-consistent noising induces more consistent SDS gradient across nearby viewpoints, whose enhanced consistency allows for coherent geometry.

interpolation (*e.g.*, bilinear, nearest neighborhood) involved in the warping process harms these properties. To overcome this issue, the warping method proposed by Chang et al. (2024) interprets a noise map as the integral of conditionally upsampled higher-resolution noise map and achieves ideal noise warping through integral noising; however, this warping process is computationally intensive, making it impractical for SDS, as it needs to be performed at each iteration.

To address this, we introduce 3D consistent integral noising, which satisfies the above criteria by utilizing an intermediate 3D point cloud representation, incorporated with conditional noise upsampling and discrete noise integral (Chang et al., 2024). We adopt 3DGS (Kerbl et al., 2023) as our 3D representation, as the mean locations of the 3D Gaussians easily define a point cloud that aligned with the geometry of the 3D scene, as described in Fig. 4(b). We then imbue each point with a random noise value sampled from a normal distribution, resulting in a 3D noised point cloud $\mathbf{n}^{3D}$, which will be projected and aggregated to produce 3D-consistent 2D noise maps, described below.

**Conditionally upsampled point cloud.** We adopt the conditional upsampling proposed in Chang et al. (2024) to 3D point cloud setting, interpreting each value in 3D point as an integration of upsampled points within a partitioned volume. Assuming this volume is a spherical volume surrounding each original point in $\mathbf{n}^{3D}$, we generate an upscaled point cloud, whose locations are sampled from a Gaussian distribution centered around the original point, as described in (c) of Fig. 4. The upscaling occurs by a factor of hyperparameter $N$, meaning that $N$ points are newly sampled for each original point $n^{3D} \in \mathbf{n}^{3D}$. Assuming an original point indexed $k$, whose noise value is $n_k^{3D}$, the noise values for its $N$ upsampled points, designated $m^{3D}$, are conditionally sampled from the original point:

$$m^{3D} \sim \mathcal{N}(\bar{\boldsymbol{\mu}}, \bar{\boldsymbol{\Sigma}}), \quad \text{with} \quad \bar{\boldsymbol{\mu}} = \frac{1}{N} \sum_k n_k^{3D}, \quad \bar{\boldsymbol{\Sigma}} = \frac{1}{N} \left( \mathbf{I}_N - \frac{1}{N} \mathbf{u} \mathbf{u}^\top \right), \tag{7}$$

where $\mathbf{u} = (1, ..., 1)^\top$ whose size is $N$, $\mathbf{I}_N$ being $N \times N$ identity matrix. In implementation, this corresponds to having $N$ noise values sampled from $\mathcal{N}(0, \mathbf{I})$, removing their mean, and adding to them $n_k^{3D}/N$. This conditional sampling is conducted independently per channel of the noise map.

**Discrete noise integral.** After conditionally upsampling the point cloud, we project its points onto a pixelized grid for a given viewpoint. Since the number of projected points may differ for each pixel,

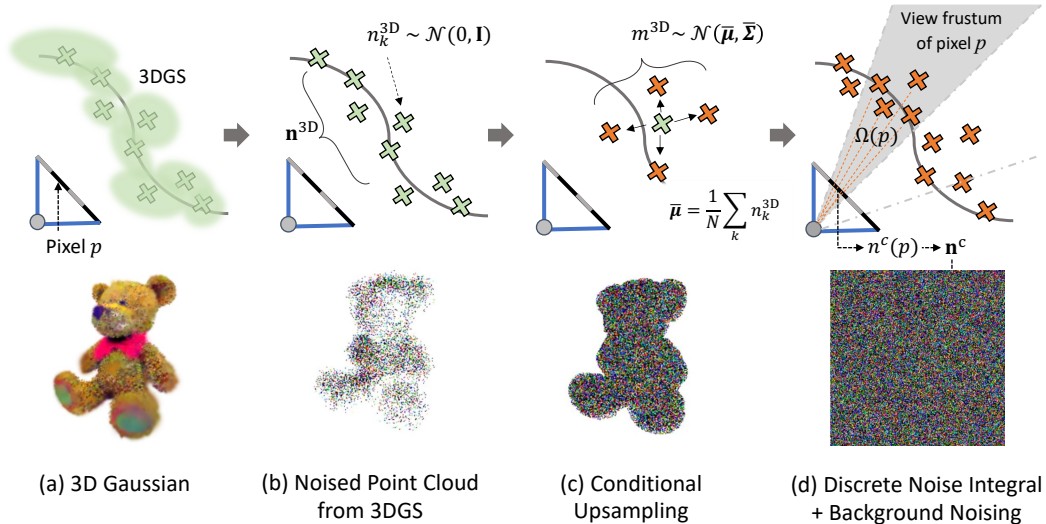

Figure 4: **3D consistent $\int$-noising.** To produce a 3D geometry-aware 2D noise map that preserves the properties of the standard Gaussian distribution, we conduct 3D conditional upsampling of point clouds and discrete integral of projected noise values. Please refer to Sec. 4.2 for more detailed explanation of the subfigures.

we perform the *discrete noise integral* to aggregate their values, obtaining a representative value for each pixel, while preserving the overall Gaussian properties of the noise map. We denote the set of noise values $m^{3D}$ of the projected upsampled noise points $\mathbf{m}^{3D}$, projected to a pixel $p$ at viewpoint $\pi$, as $\Omega(p)$. Our discrete noise is pixelwisely aggregated, summed and normalized to preserve the Gaussian properties of the noise map:

$$n^{\mathrm{c}}(p) = \frac{1}{\sqrt{|\Omega(p)|}} \sum_{m^{3D} \in \Omega(p)} m^{3D}, \tag{8}$$

where $n^{\mathrm{c}}(p)$ stands for the final, aggregated noise value for the pixel $p$ at camera $\pi$, with $|\Omega(p)|$ being the size of the set, *i.e.*, the total number of points projected to the pixel $p$. The points have no volumes forcing that each point is projected to a single pixel, which allows the integral process to occur discretely and guarantees the complete independence of pixels.

**3D consistent noises and gradients.**    Our final gradient map for viewpoint $\pi$ is defined as:

$$\mathbf{g}_{\boldsymbol{\theta},\pi}^{\mathrm{c}} = \frac{\mathcal{D}(\mathbf{z}_{\boldsymbol{\theta},\pi} + \sigma_t \mathbf{n}^{\mathrm{c}}; \sigma_t, y) - (\mathbf{z}_{\boldsymbol{\theta},\pi} + \sigma_t \mathbf{n}^{\mathrm{c}})}{\sigma_t^2}, \tag{9}$$

replacing $\mathcal{W}_{j \to i}(\mathbf{n}_j)$ with $\mathbf{n}^{\mathrm{c}}$ in Eq. 5. Our full 3D-consistency-aware SDS equation is defined as:

$$\nabla_{\boldsymbol{\theta}} \mathcal{L}_{\mathrm{SDS}}^{\mathrm{c}} \approx \mathbb{E}_{\pi \sim \Pi, \mathbf{n}^{\mathrm{c}} \sim \mathcal{N}(0,\mathbf{I}),t} \left[ \frac{\mathcal{D}(\mathbf{z}_{\boldsymbol{\theta},\pi} + \sigma_t \mathbf{n}^{\mathrm{c}}; \sigma_t, y) - \mathbf{z}_{\boldsymbol{\theta},\pi}}{\sigma_t^2} \cdot \frac{\partial \mathbf{z}_{\boldsymbol{\theta},\pi}}{\partial \boldsymbol{\theta}} \right]. \tag{10}$$

Our results in Sec. 5.2 show that our 3D consistent noising brings clear improvements to the overall quality and convergence speed of the optimization process. As hypothesized, giving 3D-geometry-aware noise to corresponding pixels in different viewpoints facilitates their SDS gradients to be more consistent, leading to faster convergence and more high-fidelity generation results.

To make the 2D noises aligned solely with the rendered surfaces, we take into account only the points that lie within a certain spherical distance from the rendered depth, preventing self-occluded surfaces from the other side of the object from influencing the noise integral process.

**Analysis.**    The validity of our method is demonstrated in Fig. 5, where we compare the 3D-consistent noise $\mathbf{n}_i^{\mathrm{c}}$ at pose $\pi_i$ produced by our method with other methods, such as warping and random noising. To this end, we compute the covariance of the produced noise, its cross-covariance with the noise of nearby viewpoint $\mathbf{n}_j^{\mathrm{c}}$ at pose $\pi_j$, and the distribution of the generated noise values. Random noising

(a) shows no correlation with nearby viewpoints, while the distributions of bilinear warping (b) and nearest warping (c) show discrepancies with standard normal distribution, with (b) especially lacking the i.i.d characteristic, as shown in the covariance matrix. 2D integral noising (Chang et al., 2024) (d) is accurate, but its heavy computation limits its usage for SDS, as the warping process must occur multiple times within a single iteration. Our method preserves the Gaussian properties such as mean, variance, and its i.i.d nature, as well as accurately representing the interpolative correlation between viewpoints, resulting an ideal 3D-consistent noise map, while computationally efficient.

### 4.3 GEOMETRY-BASED GRADIENT WARPING

To strengthen the multi-view consistency between SDS gradients during the optimization process, we introduce an additional loss based on 3D-consistent noising. This considers a mapping between 3D-corresponding locations across different viewpoints, allowing the comparison of gradients generated from distinct viewpoints. Using depth information from the rendered 3D scene (in our case, the 3DGS baseline),

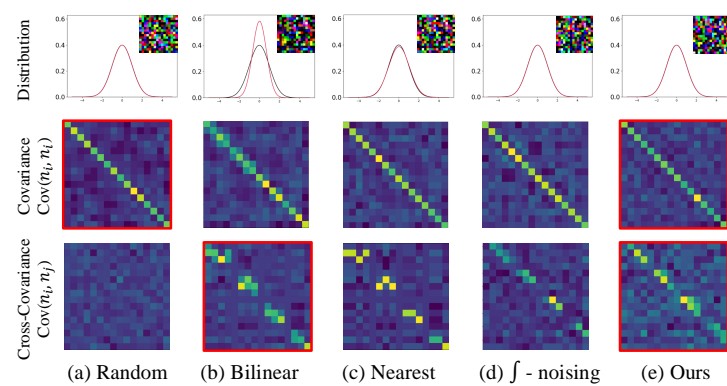

Figure 5: **Gaussian properties.** Our 3D $\int$-noising preserves the properties of standard Gaussian distribution while remaining 3D consistent.

the 2D gradient map of one viewpoint is geometrically warped to another, ensuring consistency. Specifically, the depth map, $\mathbf{d}$, helps establish pixel correspondences between viewpoints, enabling the warping of gradient maps between two viewpoints, denoted as $\mathbf{g}_1^c$ and $\mathbf{g}_2^c$.

Given two viewpoints, $\pi_1$ and $\pi_2$, and the transformation matrix $R_{1\to2}$, the corresponding pixel location $p_{1\to2}$ in $\mathbf{g}_2^c$ is calculated using the rendered depth $\mathbf{d}_1$ and the intrinsic matrix $K$. This forms a 3D geometry-based mapping function $\mathcal{W}_{1\to2}(\cdot)$, which contains correspondence information between the pixels of viewpoint $\pi_1$ and $\mathbf{g}_2^c$. By applying this mapping, the warped gradient map $\mathbf{g}_{2\to1}^c$ is generated using a nearest sampling operator, ensuring geometric consistency between viewpoints. The warping process is formalized as: $\mathbf{g}_{2\to1}^c(p_1) = \text{sampler}(\mathbf{g}_2^c; \mathcal{W}_{1\to2}(p_1))$.

### 4.4 CORRESPONDENCE-AWARE GRADIENT CONSISTENCY LOSS

We introduce **correspondence-aware gradient consistency loss**, where we penalize the dissimilarity between the gradients that have a 3D-correspondence mapping to guide the scene toward a more robust and consistent appearance and geometry. The motivation for such a loss is intuitive. Equation 6 shows that using 3D consistent noise removes much of the randomness that the noising process brought upon the SDS process, which in turn indicates that the differences between generated gradients are predominantly caused by variations in appearance and geometry.

As we are comparing the gradients generated from nearby viewpoints with nearby camera pose differences, heavy differences between corresponding gradients are highly likely to be caused by a sharp change in appearance or geometry. These sharp changes can generally be attributed to artifacts (Kwak et al., 2023; Kim et al., 2022) and geometrically inconsistent features, such as Janus problems, produced on the 3D scene. In this light, a similarity loss that forces the corresponding gradients to be more similar to one another has a regularizing effect.

Let us assume we have a gradient map $\mathbf{g}_i^c$ at the viewpoint $\pi_i$ and a warped gradient map $\mathbf{g}_{j\to i}^c$ from the viewpoint $\pi_j$. Because $\mathbf{g}_{j\to i}^c$ has been warped according to 3D geometry, the consistency loss between two adjacent viewpoints $\pi_i$ and $\pi_j$, in which $\mathbf{g}_j^c$ has been warped to $\pi_i$, is defined as follows:

$$\mathcal{L}_{\text{sim}} := \sum_{\pi_i \in \Pi} \sum_{\pi_j \in \Pi_{i,j}} \sum_p \mathbf{o}_{j\to i}(p) \cdot \left(1 - \frac{\mathbf{g}_i^c(p) \cdot \mathbf{g}_{j\to i}^c(p)}{\|\mathbf{g}_i^c(p)\|\|\mathbf{g}_{j\to i}^c(p)\|}\right), \tag{11}$$

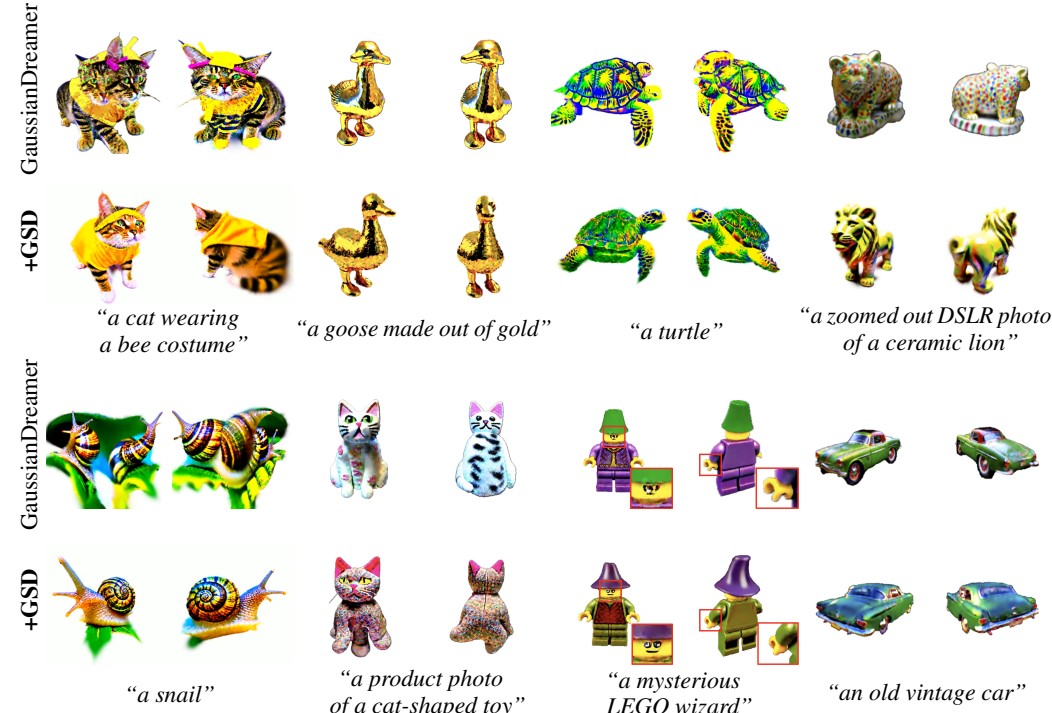

Figure 6: **Qualitative improvement over GaussianDreamer (Yi et al., 2023) baseline.** The incorporation of **GSD** framework enhances the 3D consistency generated scenes.

where $\mathbf{o}_{j \to i}$ stands for self-occlusion mask adopted from (Kwak et al., 2023), which masks out erroneously warped locations at $\mathbf{g}^{c}_{j \to i}$. Note that we back-propagate this loss only to the rendered depth $\mathbf{d}$ which was used in warping image $\mathbf{g}^{c}_{j}$ to $\pi_i$, as this loss is essentially a geometry regularizing loss. Our experimental result at 5.3 demonstrates the effectiveness of our loss in reducing geometric inconsistencies as well as aiding the generation of more fine-detailed geometry, and also shows that our loss must be used in conjunction with 3D consistent noising for proper effectiveness.

## 5 EXPERIMENTS

### 5.1 IMPLEMENTATION DETAILS

We have implemented our method using the PyTorch framework, and all our experiments were conducted with the Stable Diffusion model based on LDM (Rombach et al., 2022). The majority of our implementations were conducted on the Threestudio (Guo et al., 2023) baseline of Gaussian-Dreamer (Yi et al., 2023), and we utilized the off-the-shelf Point-E (Nichol et al., 2022) module to obtain the initial point cloud for 3D Gaussian Splatting (Kerbl et al., 2023). Our noised point cloud upsampling ratio $N = 9$, and for each iteration of the optimization process, we render batches of images separated by $5°$ from each other for consistent noising and gradient modeling.

### 5.2 QUALITATIVE ANALYSIS

Fig 6 shows the improvement that **GSD** brings to its baseline model, which is the Threestudio (Guo et al., 2023)-based GaussianDreamer (Tang et al., 2024) model. We demonstrate that our method counters such errors and geometric inconsistencies successfully, reducing multi-faced Janus problems drastically as well as fixing incoherent geometries such as multiple beaks on *"a goose made out of gold"* or two heads appearing on *"a turtle."*

In addition, in Fig. 7, to show our method's universal effectiveness across different SDS-based methodologies, we combine **GSD** with an Instant-NGP (Müller et al., 2022) based method, Prolific-Dreamer, (Wang et al., 2023) and observe the effects. As our methodology requires a point cloud aligned with scene geometry, we leverage depth map rendered via volumetric rendering to acquire the point cloud at every iteration, eliminating dependencies on external models (such as Point-E) or

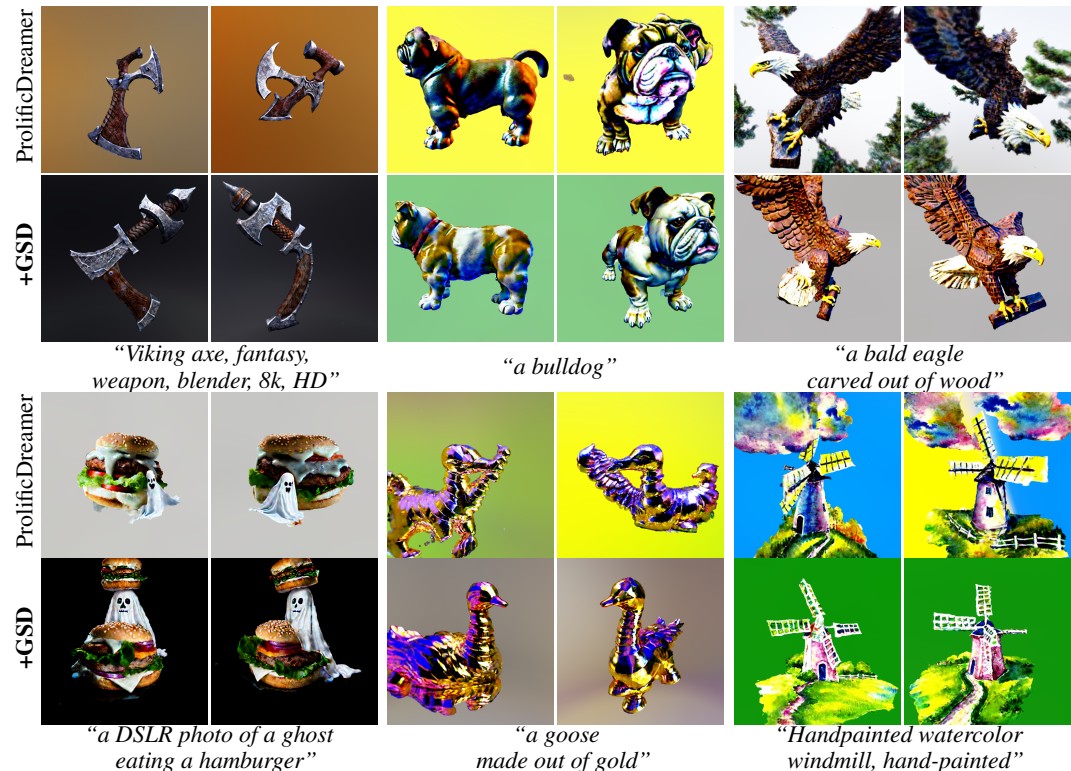

*"Viking axe, fantasy, weapon, blender, 8k, HD"*  *"a bulldog"*  *"a bald eagle carved out of wood"*

*"a DSLR photo of a ghost eating a hamburger"*  *"a goose made out of gold"*  *"Handpainted watercolor windmill, hand-painted"*

Figure 7: **Qualitative improvement over ProlificDreamer (Wang et al., 2023) baseline.** To demonstrate the effectiveness of our approach on other SDS methodologies, we apply to Prolific-Dreamer (Wang et al., 2023). Even without external models (Point-E, 3DFuse) or initializing shapes, our method improves upon overall generation, reducing various view inconsistencies and artifacts.

shape initializations. The results demonstrate that application of our approach reduces artifacts and Janus problems even in such settings. We provide more extensive experiments on other baselines in Fig. 13 which is located at our Appendix D.

## 5.3 ABLATION STUDY AND ANALYSIS

**Enhancement in convergence speed.** Our method shows effectiveness in improving the speed of SDS convergence. As shown in Fig. 9, adjusting the noise sampling strategy and applying gradient consistency loss leads to faster optimization. For example, hats and faces appear earlier when generating a "full body of a cat with a hat." This supports our approach of aligning noise maps with 3D geometry, which produces more consistent gradients and accelerates 3D representation convergence.

**Ablation on 3D consistent noising and gradient consistency loss.** We conduct an ablation study regarding our 3D consistent noising and the gradient consistency loss in Fig. 8. Our

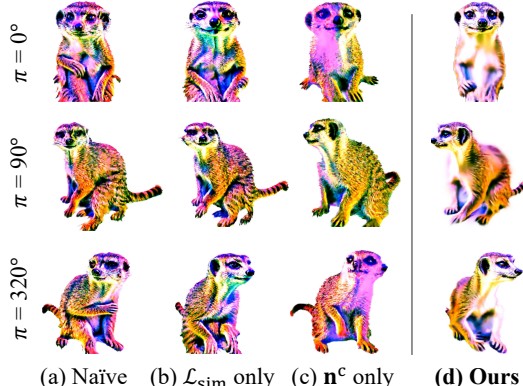

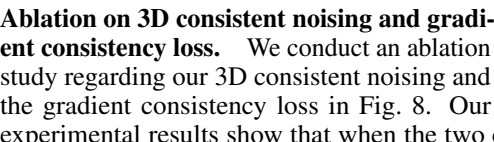
(a) Naïve  (b) $\mathcal{L}_{\text{sim}}$ only  (c) $\mathbf{n}^c$ only  **(d) Ours**

Figure 8: **Ablation.** Our experiments show that without 3D consistent noising, our consistency loss shows little to no effect on the generation process. The prompt is a *"a cute meercat"*.

experimental results show that when the two components are used in conjunction, it brings about enhancement in geometric robustness and increased fidelity from the naïve result (a), as clearly shown in (d). However, when the consistency loss is used without consistent noising, its effects are diminished, as shown in (b). Sole usage of 3D consistent noise $\mathbf{n}^c$ brings about only limited improvement as well, observable in (c). This indicates that gradient similarity incurred by 3D consistent noising is

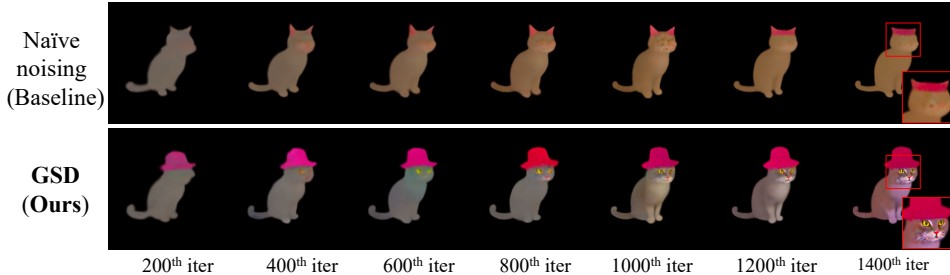

Figure 9: **Convergence speed comparison.** Comparison between naïve noising and 3D-aware noising shows that our method of 3D consistent noising and similarity loss achieves quicker convergence over baseline, GaussianDreamer (Yi et al., 2023). The prompt *"a full body of a cat with a hat"* is used.

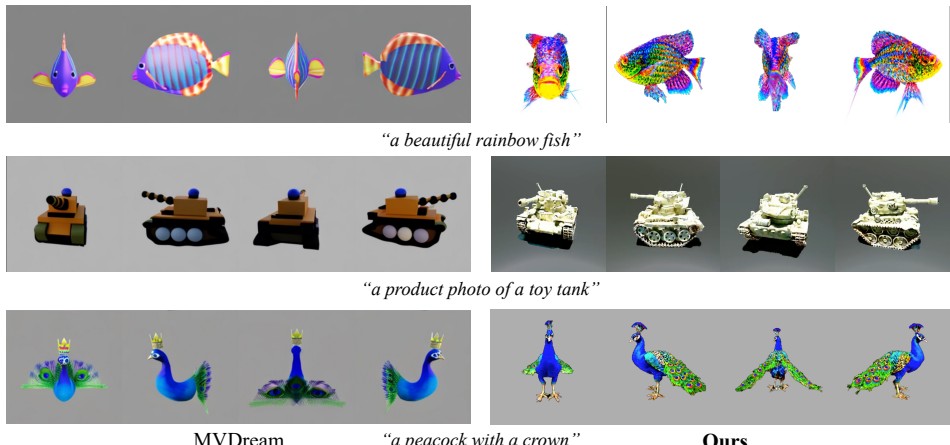

Figure 10: **Comparison to MVDream.** Generation results of **GSD**-combined SDS / VSD (Wang et al., 2023) baseline shows superior textural and geometric details in comparison to multiview generation models such as MVDream (Shi et al., 2023) given above.

crucial for gradient consistency modeling in allowing meaningful geometry regularization to take place with consistency loss.

**Comparison to multiview generation model.** We compare our framework with MVDream, a multiview generation model fine-tuned on Objaverse, which generates faster and avoids view inconsistencies. However, such fine-tuning on Objaverse, which is limited in diversity and quality of its 3D assets, causes its generation results to be constrained by having claylike, low-fidelity textures, as demonstrated in Fig. 10. In contrast, **GSD** combined with SDS methods produces detailed, high-fidelity scenes with strong geometric consistency.

**User study.** In a user study with 39 participants (Tab. 1), six multiview renderings from GaussianDreamer and ProlificDreamer were compared to **GSD**-combined results. Participants evaluated three aspects: realistic 3D geometry, adherence to the prompt, and overall quality. The results show a clear preference for **GSD**, demonstrating significant improvements.

Table 1: **User study.** The user study is conducted by surveying 39 participants to evaluate 3D coherence, prompt adherence, and rendering quality.

| Method | 3D coherence | Prompt adherence | Overall quality |
|---|---|---|---|
| Baseline + **GSD** (Ours) | **65.4%** | **68.4%** | **61.5%** |
| Baseline | 34.6 % | 31.6 % | 38.4 % |

## 6 CONCLUSION

Our method, **GSD**, integrates geometry-based correspondence into the SDS process, improving multiview consistency and geometric fidelity in text-to-3D generation. Through 3D consistent noising, gradient warping, and a multiview consistency loss, we address geometric inconsistencies without extra training or modules. **GSD** achieves competitive results and is validated by an ablation study, confirming its effectiveness in enhancing SDS-based 3D generation.

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

## A    SPHERICAL BACKGROUND NOISING

We generate 2D noise maps for the foreground and background separately and combine them to gain the final noise map, as described in Fig 2. For the foreground process, to make the 2D noises aligned solely with the rendered surfaces, we take into account only the points that lie within a certain euclidean distance, or points belonging to radius neighbor, from the rendered surface depth, preventing self-occluded surfaces from the other side of the object from influencing the noise integral process. For the background, we create a spherical point cloud surrounding the scene, which we noise, upscale, and integrate likewise, and add this noise to the empty regions of the foreground noise to produce a final, full 2D noise map retaining standard normal distribution properties.

## B    PSEUDOCODE ALGORITHM OF 3D CONSISTENT NOISING

---
**Algorithm 1** 3D Consistent Noising Process
---
1: **if** consistent_noise $=$ True **then**
2:      Configure rasterization: image_size $\leftarrow R$, point sampling radius $\leftarrow r_{\text{surf}}$
3:      Extract points: $\mathbf{P} \leftarrow$ Tensor(original point cloud)
4:      **if** $\mathbf{P} = \emptyset$ **then**
5:          Generate random tensors:
6:          $\mathbf{N} \sim \mathcal{N}(0, 1)^{(n, c_{\text{noise}})}, \mathbf{L}_{\text{rand}}, \mathbf{F}_{\text{rand}}$
7:          Upscale foreground points and features: $(\mathbf{P}_{\text{noise}}, \mathbf{V}_{\text{noise}}) \leftarrow$ NOISEUPSCALER$(\mathbf{P}, \mathbf{N})$
8:          Compute depth map: $\mathbf{D} \leftarrow$ RENDERDEPTH$(\mathbf{P})$
9:          Project foreground noise to 2D: $\mathbf{M}_{\text{fore}} \leftarrow$ REPROJECTOR$(\mathbf{P}_{\text{noise}}, \mathbf{V}_{\text{noise}}, \mathbf{D})$
10:          **if** background $=$ True **then**
11:              Generate background noise: $(\mathbf{P}_{\text{bg}}, \mathbf{V}_{\text{bg}}) \leftarrow$ SPHERENOISE$()$
12:              Project background noise to 2D: $\mathbf{M}_{\text{bg}} \leftarrow$ REPROJECTOR$(\mathbf{P}_{\text{bg}}, \mathbf{V}_{\text{bg}})$
13:          **end if**
14:      **end if**
15:      Add foreground and background noise map: $\mathbf{M}_{\text{noise}} \leftarrow \mathbf{M}_{\text{fore}} + (\mathbf{M}_{\text{fore}} = 0) \cdot \mathbf{M}_{\text{bg}}$
16: **end if**
---

## C    EMULATING THE JANUS PROBLEM IN 2D SCORE DISTILLATION SAMPLING

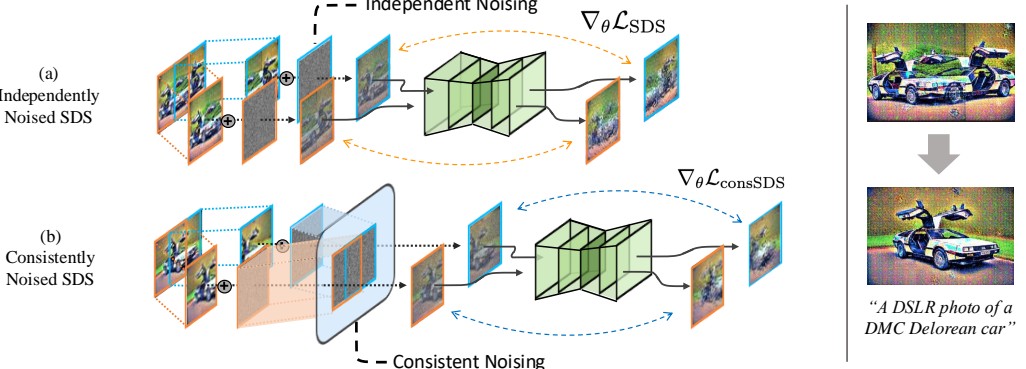

Figure 11: **Design for consistent noising experiment in 2D SDS.** To observe the effects of consistent noising within SDS process, we design an experiment which compares the generation results of panoramic image generated from independently-noised 2D SDS process with that of consistently-noised 2D SDS process.

To verify our hypothesis that the key reason for the Janus problems in text-to-3D generation is inconsistent gradient between different viewpoints, we design a toy experiment in a simplified setting, which is SDS-based optimization of 2D image pixels similar to (Hertz et al., 2023). Our objective is to observe the effect that such noise consistency between SDS processes induces upon the score distillation process. As described in Figure 11, we optimize a rectangular, panorama-shaped 2D tensor by cropping it into multiple square subsections and applying SDS on each crop. This setting bears a strong analogy to text-to-3D optimization in that a single global representation is cropped into multiple subsections, which are optimized separately via score distillation.

First, let us assume a naïve setting in which all subsections are passed through independent diffusion processes, in which the correspondences within overlapping areas of neighboring subsections are completely ignored. This setting, as shown in the top row of Fig 12, results in a broken image with each crop containing separate, inconsistent generations, displaying difficulty in optimizing the overlapping region and failing to achieve global coherency. Notice that this phenomenon closely resembles the Janus problem occurring in text-to-3D, with multiple faces appearing across the crops. It also clearly demonstrates how giving consistent noise to overlapped regions largely removes these effects, allowing coherent single scene to emerge across different cropped windows.

This "2D version of the Janus problem" shown in the toy experiment strengthens the hypothesis that the culprit behind the Janus problem is indeed the absence of correspondence awareness in the current SDS formulation, and how it can be largely resolved simply by applying consistent noising to the overlapped regions.

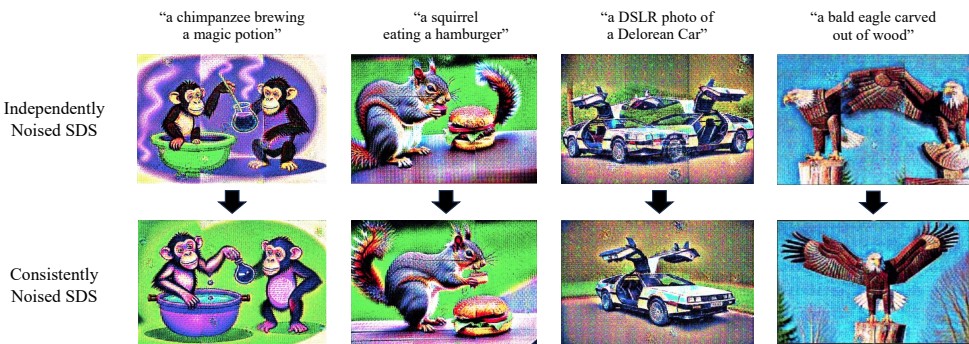

Figure 12: **Effect of consistent noising in 2D SDS.** In the top row, where all subsections (windows) are passed through independent diffusion processes, Janus-like effect in 2D panoramic image occurs, showing multi-faced artifacts in different sections of the image. When consistency between noise between the windows are introduced, it can be seen that overall consistent image is generated.

## D   ADDITIONAL RESULTS ACROSS VARIOUS BASELINES

In Fig. 13, to show our method's universal effectiveness across various SDS-based methodologies, we combine **GSD** with other Instant-NGP Müller et al. (2022) based baseline methods Poole et al. (2023); Wang et al. (2023) and observe the effects. As our methodology requires a point cloud aligned with scene geometry, we leverage 3DFuse Seo et al. (2024), which conditions scene optimization on a point cloud. As the generated scene geometry closely follows the point cloud, we leverage this point cloud to conduct 3D consistent integral noising. Our results reveal that despite using 3DFuse, which is designed to enhance view consistency of generated 3D scenes, artifacts and view inconsistency problems such as the Janus problem persist in numerous generated results. Application of our approach brings about clear enhancements in these aspects, resulting in more geometrically robust and well-textured 3D scenes.

## E   ADDITIONAL COMPARISON TO PREVIOUS WORKS

In Fig. 14, we compare the performance of our method to other baseline methods Poole et al. (2023); Wang et al. (2023); Yi et al. (2023); Seo et al. (2024). Other approaches are shown to yield

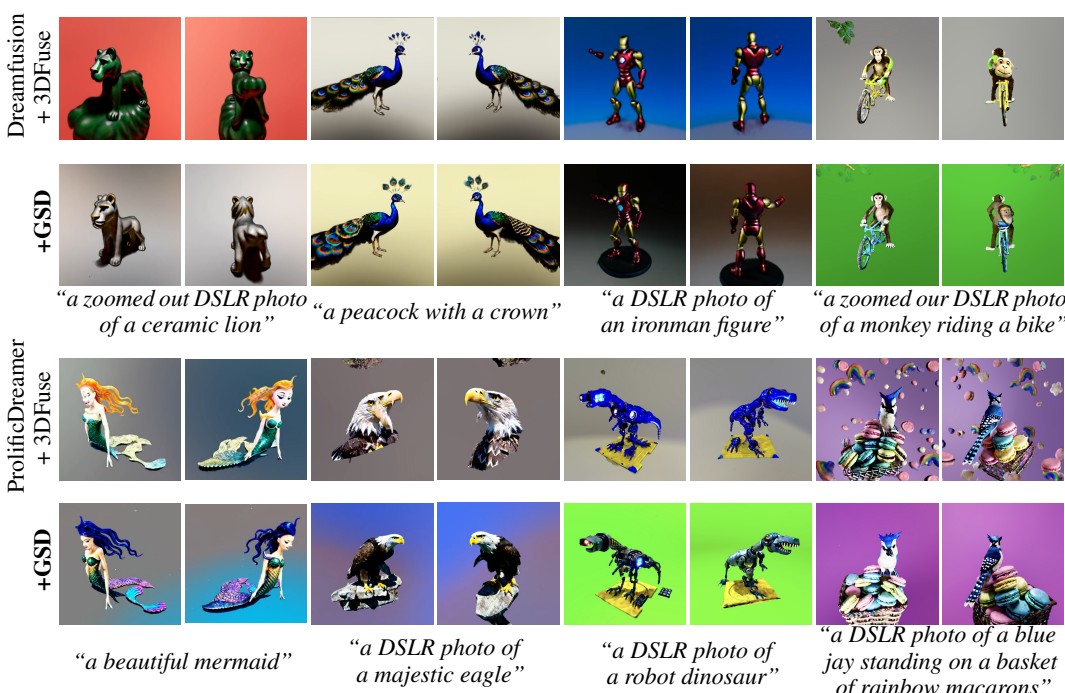

Figure 13: **Qualitative improvement over Dreamfusion (Poole et al., 2023) and Prolific-Dreamer (Wang et al., 2023) baselines combined with 3DFuse (Seo et al., 2024).** To demonstrate the effectiveness of our approach on other SDS methodologies, we apply **GSD** to 3DFuse (Seo et al., 2024)-combined Dreamfusion (Poole et al., 2023) and ProlificDreamer (Wang et al., 2023). Our method improves upon overall generation, reducing various view inconsistencies and artifacts.

inconsistent, distorted geometries across multiple directions, or undergo the Janus problem, producing features that should be seen at the front in other viewpoints of the scene. Erroneous markings on the texture can also be observed. Our approach, however, displays robustness regarding both geometric consistency and texture of the scene, as demonstrated by the given figures.

## F    ANALYSIS IN COMPARISON TO MULTIVIEW GENERATION MODEL

In Fig. 10, we compare the generation results of our framework with MVDream Shi et al. (2023), a multiview generation diffusion model fine-tuned on a 3D dataset, Objaverse Deitke et al. (2023). This family of text-to-3D generation models Liu et al. (2023); Shi et al. (2023) is capable of directly predicting novel viewpoints of a given image or text, allowing for faster generation speed that SDS-based frameworks, with MVDream nearly completely free from view inconsistency problems. However, such fine-tuning on Objaverse, which is limited in diversity and quality of its 3D assets, causes its generation results to be constrained by having claylike, low-fidelity textures. In comparison, we show that **GSD** combined with SDS methodologies (GaussianDreamer and ProlificDreamer in given results) is capable of creating scenes of highly detailed geometry and fidelity, fully leveraging the generative capability of a pretrained 2D diffusion that has not been fine-tuned to Objaverse, while also demonstrating strong geometric robustness and consistency as our **GSD** encourages view-consistent generation through score distillation process itself.

## G    360° VISUALIZATION OF 3D SCENE AND CONSISTENT NOISE

Fig. 15 displays a 360° comparison of our methodology with that of baseline, which shows drastic improvement induced by the application of **GSD**. The experiment shows an interesting case demonstrating how our method functions: even though the conditioning geometry is completely identical due to constraint by 3DFuse, the incorporation of our methodology encourages a more view-consistent and realistic interpretation of this given geometry, outputting a drastically enhanced 3D scene optimization result, as well displayed.

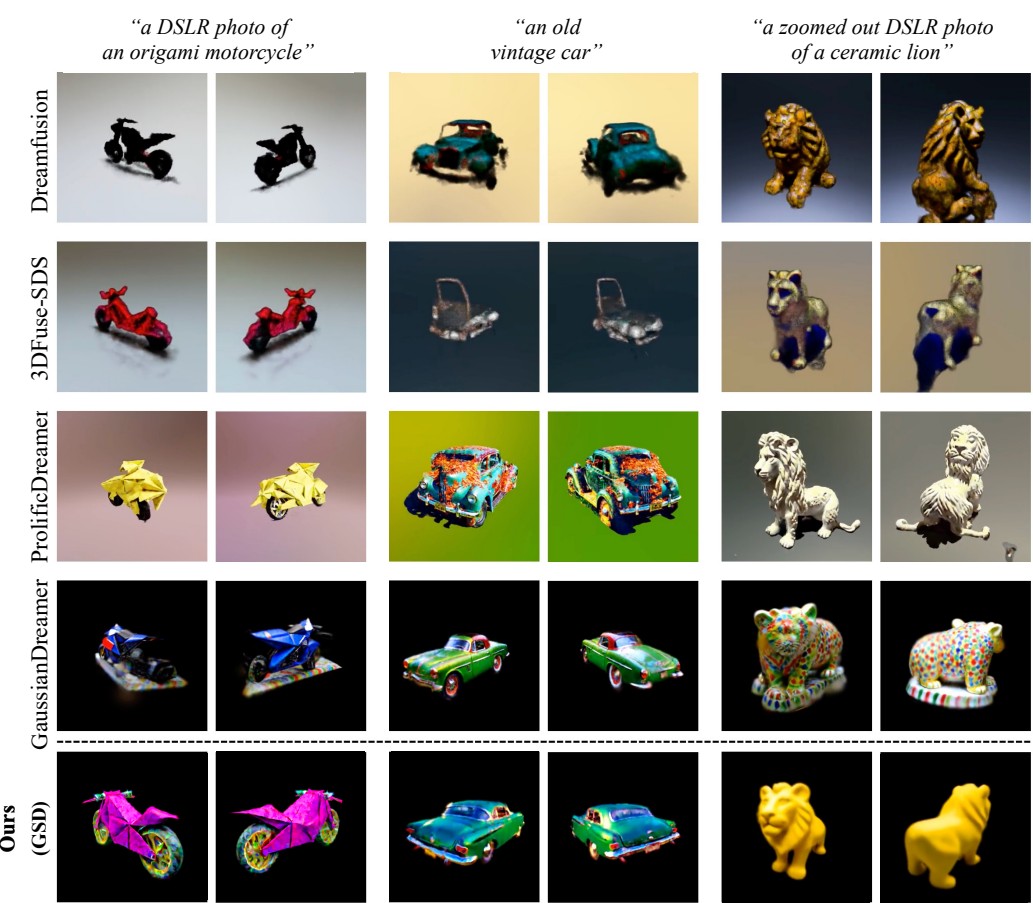

Figure 14: **Comparison to previous works.** We compare our framework with other text-to-3D frameworks: DreamFusion Poole et al. (2023), ProlificDreamer Wang et al. (2023) and Gaussian-Dreamer Yi et al. (2023). Our method achieves more geometrically consistent results while being closely faithful to the text prompt given, demonstrating its effectiveness and stability.

## H CONSISTENCY ANALYSIS WITH CLIP SIMILARITY

To measure the consistency of generated 3D objects, we follow previous work (Hong et al., 2023) of measuring the CLIP similarity between the generated images and the front view and back view prompts across various prompts and provide the result at Fig. 16. However, we do not find a significant correlation between the view prompts and the images corresponding to each view. This appears to be partially because the CLIP model, being discriminative, does not accurately evaluate the similarity between detailed prompts and images.

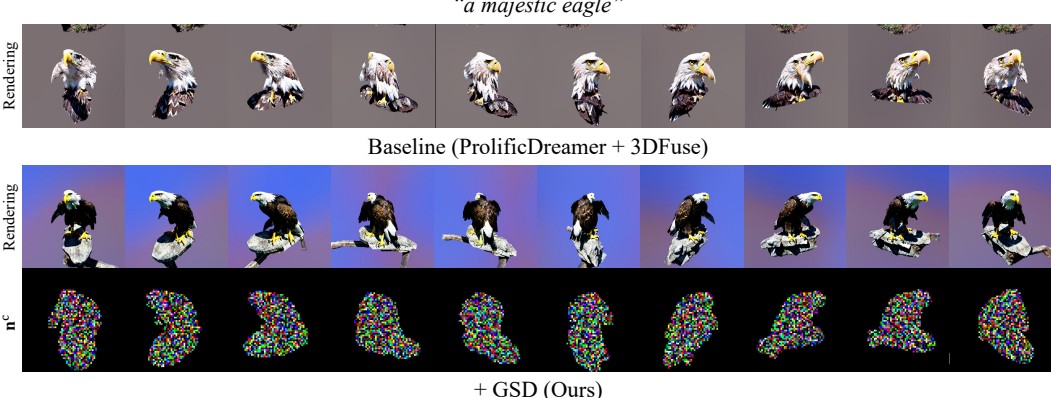

Figure 15: **360° visualization.** 360° comparison between the baseline model, which is Prolific-Dreamer (Wang et al., 2023) combined with 3DFuse (Seo et al., 2024), and results with **GSD** added, along with 3D consistent noise visualization.

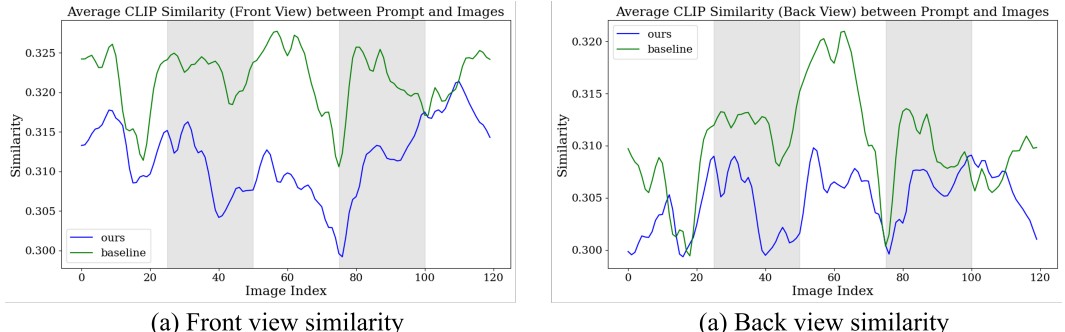

(a) Front view similarity          (a) Back view similarity

Figure 16: **CLIP similarities between each rendered image and view-augmented prompt and images.** We compute CLIP similarities for each image and view-augmented prompt (e.g., *"front view of"* and *"back view of"*). The x-axis value (image index) corresponds to the azimuth, where 0 stands for the front view and 60 for the back view. The baseline used for this experiment is GaussianDreamer (Yi et al., 2023).