# OpenReview forum: "Geometry-aware Score Distillation via 3D Consistent Noising and Gradients"
_ICLR.cc/2025/Conference — Submitted to ICLR 2025_

### Official Review · Reviewer_Wsmb · 2024-10-19

**Soundness:** 2
**Presentation:** 2
**Contribution:** 2
**Rating:** 5
**Confidence:** 4

**Summary:**

The authors propose a method to enhance SDS-based 3D generation through 3D consistent noising and geometry-aware gradient warping. The authors presented an algorithm that could warp the gaussian noise at different camera view while maintaining Gaussian properity for one camera view. The 3D consistent noising aims to improve gradient consistency, while the geometry-aware gradient warping seeks to reduce geometric artifacts.

**Strengths:**

1. The proposed noising method establishes a correlation between camera views. The noising methods can generate i.i.d. Gassian noise correctly while maintaining correlation properities.

2. The introduction of the *correspondence-aware gradient consistency loss* serves as a regularization technique to mitigate geometry artifacts.

**Weaknesses:**

1. The motivation behind *3D consistent noising* lacks clarity. Although the authors reference its similarity to $\int$-noise [1] for improving texture consistency in video generation, the discussion on its application in 3D generation is insufficient. This omission raises questions about how 3D consistent noising can enhance 3D generation specifically. The introduction of 3D consistent noising appears unrelated to the 3D score interpretation of SJC, raising concerns about its relevance and application within this context. Thus, I suggest the authors to provide a more detailed explanation of how 3D consistent noising specifically addresses challenges in 3D generation.

2. The noising method is confined to point cloud (3D GS) representation, limiting its applicability to other forms such as NeRF or Mesh. Additionally, the algorithm appears to overlook occlusion; for instance, a Gaussian particle not visible from the current camera view may still influence noise computation, resulting in questionable correlations compared to surface warp (optical flow) as in $\int$-noise [1].

3. The experimental evaluation is weak, with qualititive results from both the baselines and proposed methods showing low quality (Fig. 6, 7, 11). The results strongly underperform SoTA implementations like ProlificDreamer (VSD) and LucidDreamer (ISM), undermining the validity of the user study. Furthermore, this paper is lack of quantitative metrics. For the CLIP score in Fig. 13, details on the number of prompts and seeds used in the experiments are missing, and a more thorough experimentation (with at least 25 prompts) is needed.

4. The *correspondence-aware gradient consistency loss* remains difficult to grasp despite the explanations in Sec. 4.4. The rationale behind why sharp changes are likely artifacts needs clarification. I suggest the authors to provide a more intuitive explanation of why sharp changes are likely to be artifacts. Additionally, the ablation study in Fig. 8 fails to effectively illustrate its impact; more results with varied seeds or prompts would strengthen the argument.

**Questions:**

1. I recommend including a pseudo-algorithm to enhance comprehension. Is the noise regenerated after each batch of gradient updates? Does the computation of correlation occur solely within the same batch?

2. Are noise particles not visible considered in the computation? If so, why not focus on visible particles, aligning more closely with the computation in $\int$-noise [1]?

3. There are missing experimental details as noted in weakness 3. How many prompts do you experimented with?

[1] How I Warped Your Noise: a Temporally-Correlated Noise Prior for Diffusion Models (https://warpyournoise.github.io/)

---

> ### Author Response · Authors · 2024-11-28
> **Response to Reviewer Wsmb**
>
> **W1. The motivation behind 3D-consistent noising lacks clarity.**
>
> **A1. Clarification on 3D-consistent noising.**
>
> We argue that 3D-consistent noising significantly enhances 3D generation by enforcing consistency among 2D scores distilled from different viewpoints. Our analysis identifies the root cause of the geometric inconsistency problem, commonly referred to as the Janus problem, in text-to-3D generation as the **lack of gradient coherence across viewpoints**. By addressing this fundamental issue, 3D-consistent noising ensures a more unified and accurate optimization process, improving multiview consistency in the generated 3D models.
>
> To investigate whether geometric inconsistencies stem from a lack of consistency across different viewpoints and to evaluate if providing consistent noise to overlapping regions can mitigate this issue, **we demonstrate an additional toy experiment in Appendix C of our revised paper**. In this experiment described in Figure 11 and 12, we optimized a panoramic-resolution tensor (H, W, 3), divided into multiple windows, with each window independently subjected to SDS optimization for text-to-image generation. The results reveal that when noise was sampled independently for each viewpoint, the SDS-optimized image exhibited several artifacts, such as an eagle with multiple heads or two cars appearing on opposite sides of the image—issues resembling the Janus problem observed in 3D generation. However, when overlapping regions were assigned consistent noise across different windows, these artifacts were significantly reduced, resulting in a more coherent and consistent image across the entire panorama. These findings highlight the pivotal role of consistent noise in enhancing multiview coherence and reducing artifacts in SDS-based optimization.
>
> This 2D experiment provides a compelling analogy for the SDS-based text-to-3D generation problem we aim to address. In this context, the panoramic image represents the full 3D structure, which cannot be entirely observed from a single viewpoint, while each window corresponds to a specific view rendered from a particular camera pose. Just as consistent noise across overlapping regions in the 2D experiment leads to greater coherence, we can reasonably infer that ensuring consistent noising for overlapping 3D regions observed from multiple viewpoints can mitigate geometric inconsistencies, including the Janus problem, in the 3D domain. This analogy reinforces the importance of our approach in aligning multiview perspectives to achieve a unified and artifact-free 3D representation.
>
> &nbsp;
>
> **W2. Limitations in representation (e.g., NeRF or Mesh) and overlooking occlusion effects.**
>
> **A2-1. Generalizability to other representations (NeRF and Mesh)**
>
> First, we clarify that our noising method is **not limited to point cloud (3DGS) representation**.
> The point cloud serves only as an auxiliary 3D shape to facilitate 3D-consistent noising; the primary 3D representation that is optimized is not restricted to any specific representation.
> For this reason, Figure 7 of our initially submitted paper (Figure 13 of our revised paper), we had already demonstrated our results on Dreamfusion [1] and ProlificDreamer [2] whose 3D representation is NeRF [3] and Instant-NGP [4], respectively. In these experiments, we have used 3DFuse to align the generated 3D scene to the point cloud initially generated by Point-E [5].
>
> Addtiionally, in Figure 7 of our revised paper, we provide our additional results on ProlificDreamer baseline, **which does not rely on any initial point cloud (generated from Point-E) or 3DGS [6].** Instead, we obtain our point cloud directly from volumetric-rendered depths of given viewpoints and leverage it for our consistent noising process. This allows us to obtain point cloud aligned to 3D scene at every iteration without Point-E or initial point clouds. Our results demonstrate that even in this scenario, in which no external models (Point-E, 3DFuse) or shape initializations are used, our method effectively improves 3D geometry - showing that our noising method is not confined to 3DGS representation.
>
> &nbsp;
>
> **A2-2. Generalizability to other representations (NeRF and Mesh)**
>
> Our algorithm addresses occlusion by focusing on the points belonging to radius neighbors from the rendered depth acquired through conducting volumetric rendering of depth in the neural 3D representation. This prevents “self-occluded” surfaces from influencing the noise integral process. Note that this information was **already briefly described in the Appendix A** of the initially submitted manuscript. Following the reviewer's comment, it has been emphasized in the main paper's revised version (Section 4.2). Thank you for highlighting the importance of this clarification.

---

> ### Author Response · Authors · 2024-11-28
> **Response to Reviewer Wsmb**
>
> **W3. Weak experimental evaluation**
>
> **A3. Improved experimental evaluation in the revised paper.**
>
> In Figure 7 of the revised version of our paper, we have included new qualitative results with additional prompts which shows **generation qualities on par with ProlificDreamer**, while also demonstrating our model's effectiveness in scenarios where initial point cloud or external models are not present. We also included a **pseudo-code describing our consistent noising process** in Appendix B. All experiments show that our method effectively reduces the Janus problem compared to baselines, as shown in the revised figures. While it does not completely eliminate the problem, **as a plug-and-play method**, it offers substantial improvements without requiring additional training or 3D priors.
>
> &nbsp;
>
> **W4. Clarification required on the correspondence-aware gradient consistency loss**
>
> Gradient dissimilarity loss (Equation 11) works by back-propagating the loss evaluating 3D inconsistencies between gradient maps to the 3D scene, which optimizes the geometry and texture of the scene so that its renderings produce more 3D consistent gradients. This loss encourages the 3D scene to be optimized to obtain geometry and texture whose renderings yield consistent gradients across different viewpoints. Essentially, it acts as a regularization mechanism, promoting coherence by aligning gradients from multiple perspectives in a manner similar to the epipolar constraint.
>
> In optimized neural 3D representations, artifacts such as geometric inconsistencies often manifest as erroneous shapes that appear correct from certain viewpoints but reveal broken or distorted geometry when observed from others. These artifacts usually result from the 3D representation **erroneously overfitting** to certain viewpoints. This phenomenon is what we refer to as “sharp changes across viewpoints.” Such abrupt differences are unlikely to correspond to true geometry or texture and instead indicate errors or artifacts introduced during the optimization process. As a result, these inconsistencies become prime targets for regularization, ensuring that the 3D representation remains coherent and consistent across all viewpoints.
>
> We acknowledge the reviewer’s suggestion regarding Figure 8 and agree that additional results with varied seeds or prompts would strengthen the argument. To address this, we are conducting addition experiment to be demonstrated through anonymous Github page, and we will include the comprehensive ablation studies in the revised manuscript to better illustrate the impact of the gradient dissimilarity loss under diverse conditions. These additional results will further highlight its effectiveness in improving multiview consistency and reducing artifacts.
>
> &nbsp;
>
> [1] Poole, Ben et al. “DreamFusion: Text-to-3D using 2D Diffusion”, ICLR 2023
>
> [2] Wang, Zhenjyi et al. “ProlificDreamer: High-Fidelity and Diverse Text-to-3D Generation with Variational Score Distillation”, NeurIPS 2023
>
> [3] Mildenhall, Ben et al. “NeRF: Representing Scenes as Neural Radiance Fields for View Synthesis”, ECCV 20203
>
> [4] Muller, Thomas et al. “Instant Neural Graphics Primitives with a Multiresolution Hash Encoding”, SIGGRAH 2022
>
> [5] Nichol, Alex et al. “Point-E: A System for Generating 3D Point Clouds from Complex Prompts”, ArXiv 2022
>
> [6] Kerbl, Bernhard et al. “3D Gaussian Splattingfor Real-Time Radiance Field Rendering”, SIGGRAPH 2023

---

> > ### Comment · Reviewer_Wsmb · 2024-11-29
> >
> > I appreciate the detailed response from the authors, particularly the clarifications regarding the applicable representations and potential occlusion issues, which addressed some of my concerns. However, the experimental results remain unconvincing, with most of the outputs being of low quality. Additionally, the lack of quantitative experiments has not been sufficiently addressed. As a result, I am inclined to increase my score to 5, but I would prefer to make a final decision after reviewing the additional experiments, which are expected to be demonstrated through an anonymous GitHub page (though it seems the URL has not been provided yet).
> >
> > The authors claim that consistent noising can help reduce multi-view problems, but this has not been demonstrated effectively either theoretically or through experiments. The multi-face problem arises mainly due to the lack of positional control in the diffusion process and insufficient information from other camera views. While consistent noising may improve gradient consistency, it does not necessarily prevent the model from generating a face on the back of the object, which is central to resolving multi-face issues. As illustrated in figures (e.g., Figure 13), baseline methods do not always produce multi-faced results, and the GSD still exhibits multi-faced issues (e.g., top-right corner of Figure 13) or multiple wings (e.g., top-right of Figure 7). A quantitative evaluation of the multi-face problem, using a larger sample size (e.g., 25 prompts), would be helpful. Furthermore, shape initialization methods, such as 3D fuse, may substantially reduce this problem, as also highlighted by Reviewer cMr2.
> >
> > Additional Suggestions and Questions (not related to score):
> >
> > The authors have included a noising pseudo-algorithm, which aids in understanding the approach. However, I suggest adding an algorithm detailing the training iterations and loss computation, as this would provide additional clarity for readers.

---

> > > ### Author Response · Authors · 2024-12-03
> > > **Additional Response to Reviewer Wsmb**
> > >
> > > Thank you for your comment. First, we would like to emphasize that we do not argue that our method brings about complete elimination of the Janus problem in 3D generation. However, the core of our paper is that we identify that **a key source of the Janus problem as the lack of consistency between each viewpoints' SDS process**, demonstrated especially in Appendix C - and that it can be efficiently addressed through simply making the noising process consistent between SDS processes of different viewpoints.
> > >
> > > Our enforcing of viewpoint consistency using 3D consistent noising is akin to resolving "insufficient information from other views" that you have mentioned through modification in diffusion nosing process, as consistency induced through 3D consistent noising between different viewpoints (e.g. front and side, side and back) effectively serves as regularization throughout the optimization process, leading to better global 3D scene alignment.
> > >
> > > As stated as answer to reviewer cMr2, usage of shape initialization methods or 3DFuse does not completely mitigate this problem, but even in cases when these methods still resulted Janus problem, our model does successfully reduce this issue in regards to such cases, as demonstrated in Figure 13 of our revised paper.
> > >
> > > Lastly, please note that our method is **a plug-and-play method**, and our main advantage derives from the fact that our method requires no finetuning of the diffusion models, incorporation external models, or initial 3D shapes, but solely brings about significant reduction in geometric inconsistency only through usage of 3D consistency noising and gradient consistency loss. We argue this is an interesting finding worthy of further investigation and discussion in the field of text-to-3D generation.
> > >
> > > We will promptly follow with our video results in our anonymous Github page demonstrating our video results, showing notable reduction in Janus problem throughout the global scene, effective with only a simple incorporation of our method into existing baselines, as well as our noise visualization results.

---

### Official Review · Reviewer_C6Fp · 2024-10-28

**Soundness:** 2
**Presentation:** 3
**Contribution:** 2
**Rating:** 5
**Confidence:** 4

**Summary:**

This paper aims to address the well-known Janus problem of the SDS loss. Specifically, it proposes modifying the noise sampling process of the SDS loss by replacing random noise with 3D-consistent noise. The 3D-consistent noise is generated by extending the integral noise method of Chang et al. (2024) using an intermediate point cloud representation.

**Strengths:**

1. The paper is well-organized, easy to follow, and clearly presented.
2. The proposed method offers a fresh perspective on the multi-face Janus problem of the SDS loss, particularly from the standpoint of random noise sampling.
3. Although the solution is an extension of existing work, it is still novel in the 3D generation field.

**Weaknesses:**

1. My primary concern lies in the use of latent diffusion models, as all experiments in the paper are conducted on latent diffusion models. While I do not doubt that the proposed method could be effective on pixel-space diffusion models, the entire analysis may not be applicable to latent diffusion models.

- **First,** 3D-consistent noise in latent space does NOT equate to consistent gradients in pixel space. Imagine a simple case where the latent map is shifted to the right or left by a few pixels—the VAE would not decode the same image with the same shift in pixel space. In the case of SDS combined with latent diffusion, the gradient at the latent space needs to pass through a VAE encoder composed of multiple convolutional layers and activation functions. As a result, simply keeping the gradient of a portion of the latent map unchanged does not ensure that the corresponding image regions will also receive an unchanged gradient.

- **Second,** this limitation has already been highlighted in the original work of Chang et al. (2024) (see their Appendix E). The authors of the integral noise method in Chang et al. (2024) noted that their method does NOT perform well in latent diffusion models and provided three reasons for this. Could the authors clarify why integral noise fails in the temporal domain but succeeds in the 3D case using LDM?

2. The paper does not address the limitations of the proposed method.

**Questions:**

My major concern is the applicability of the proposed method in latent diffusion models. Please refer to the weaknesses outlined above.

**Details Of Ethics Concerns:**

N.A.

---

> ### Author Response · Authors · 2024-11-28
> **Response to Reviewer C6Fp**
>
> **W1. Consistent gradients in Latent Diffusion Models are not ensured and lack correspondence.**
>
> **A1. Noise consistency remains effective in latent diffusion models.**
>
> Despite differences between how the noise is interpreted in latent space and pixel space, it has been theoretically and empirically shown in works such as MultiDiffusion [1] that **gradient consistency in latent space does effectively translate to the final generated image at pixel space**. For example, in MultiDiffusion, a gradient manipulation method formulated in pixel space, named diffusion path fusing, is implemented in LDM (Stable Diffusion) [2] and shows clear effectiveness in latent space. This suggests that certain manipulations of gradients in the latent space can yield consistent and coherent results in pixel space. By leveraging this principle, our approach, which incorporates 3D-consistent noising and gradient dissimilarity loss, **aims to enforce multiview consistency at a semantic level**, ensuring that the improvements in latent space contribute to meaningful reductions in geometric inconsistencies and artifacts in the final 3D outputs.
>
> Furthermore, "How I Warped Your Noise'' [3] also states that *"the assumption that moving (warping) the noise helps with temporal coherency still holds in latent models''* in Appendix E.1 of their paper, showing that noise warping on LDM does show notable effects in preserving consistency between frames, despite not showing fine-grained coherency expected from pixel diffusion models. In the same section, the limitations state that “the (warped) noise used in latent diffusion models does not contribute directly to the fine details of the results in image space,'' implying that consistency holds only to high-level semantic details from temporal coherency within the latent space.
>
> As the Janus problem in text-to-3D generation mostly appears in semantic levels (such as multiple faces), we argue that semantic-level consistency from noise warping shown in LDM is sufficient to reduce geometric inconsistencies and alleviate the Janus problem. To this end, in Figure 11 and 12 of our revised paper, we demonstrate that such consistent noising is relevant even in LDM-based score distillation, through a consistent-nosing experiment conducted in 2D domain.
>
>  &nbsp;
>
> [1] Bar-Tal, Omer et al. “MultiDiffusion: Fusing Diffusion Paths for Controlled Image Generation”, ICML 2023
>
> [2] Rombach, Robin et al. “High-Resolution Image Synthesis with Latent Diffusion Models”, CVPR 2022
>
> [3] Chang, Pascal et al. “How I Warped Your Noise: a Temporally-Correlated Noise Prior for Diffusion Models”, ICLR 2024

---

> > ### Comment · Reviewer_C6Fp · 2024-11-28
> > **Reviewer comment**
> >
> > I think my concerns are not addressed well in the authors' response .
> >
> > First, let me quote the full text in Appendix E of Chang, Pascal et al:
> >
> > **Noise warping does not improve temporal coherency in latent diffusion models as much as one
> > would expect.** We pinpoint this observation to three main reasons. First, temporally coherent images
> > do not necessarily translate into temporally coherent autoencoder latent vectors. Thus, temporally
> > consistent noise priors in the VAE latent space might be suboptimal. Second, the noise used in
> > latent diffusion models does not contribute directly to the fine details of the results in image space,
> > which is where warped noise priors excel. Lastly, the autoencoder almost always introduces a nonnegligible reconstruction error that affects the temporal coherency of the results, independent of the
> > latent diffusion process.
> > From a more theoretical perspective, the temporally-correlated infinite-resolution noise warping we
> > propose is only possible because of the structural self-similarity of Gaussian noise, which allows
> > us to interpret a discrete Gaussian noise sample as an aggregated view of more Gaussian samples
> > at a smaller scale. The warping could thus be operated at the limit, in the continuous setting. This
> > core assumption does not hold in latent diffusion models, because temporal coherency is no longer
> > targeted in the latent space where the noise resides, but rather in the image space mapped by the
> > decoder. Unfortunately, Gaussian noise decoded by the VAE no longer possesses the self-similarity
> > property. This aspect can be visualized in the decoded noise shown by Figure 17a (a) (iii).
> > **Nonetheless, the assumption that moving the noise helps with temporal coherency still holds in latent
> > models to some extent**, as some of our experiments below can show.  One of our early observations
> > was that the VAE decoder is translationally equivariant in a discrete way, i.e. translating the latent
> > vector by an integer number of pixels leads to an almost perfectly shifted image when decoded. For
> > an autoencoder with a compression factor f = 8 (e.g. Stable Diffusion), this means that a 1-pixel
> > shift in the latent space would produce a 8-pixel shift in the final image.
> >
> > I think this description does not align with the core motivation of this paper. Also, in the response above, the authors removed the words **to some extent**.
> >
> > Moreover, the authors referenced MultiDiffusion. There is notable difference between the proposed method and MultiDiffusion: the proposed method requires backpropagation through the encoder, whereas MultiDiffusion does not. Additionally, I am unable to find the theoretical proof regarding gradient consistency in the MultiDiffusion paper. Could the authors clarify what is shown "theoretically" (quote).

---

> > > ### Author Response · Authors · 2024-12-03
> > > **Additional Response to Reviewer C6Fp**
> > >
> > > We thank the reviewer for their detailed response and for providing further clarification regarding their concerns. Below, we address the points raised:
> > >
> > > **On the Interpretation of Appendix E in “How I Warped Your Noise”**
> > >
> > > We appreciate the reviewer’s detailed citation of Appendix E, and we acknowledge that the temporal coherence described in this work focuses primarily on pixel-space manipulations. However, we respectfully emphasize that the limitations stated in Appendix E.1, while valid, are not in conflict with the core motivation of our paper. Specifically:
> > >
> > > - The critique regarding temporal coherence in latent diffusion models primarily addresses fine-grained coherency in pixel space. However, the Janus problem we address in our work arises at a **semantic level**, such as multi-facial inconsistencies, rather than at the level of fine-grained detail.
> > > - The quoted text also states that the assumption of noise manipulation aiding temporal consistency “still holds in latent models to some extent.” This aligns with our claim that latent-space noise manipulations, while not perfect, can enforce a meaningful degree of multiview consistency at a high-level semantic scale.
> > >
> > > We used “to some extent” selectively in our initial response to align with the semantic-level nature of the Janus problem in text-to-3D tasks, which differs fundamentally from fine-grained temporal coherence tasks in “How I Warped Your Noise.”
> > >
> > > **MultiDiffusion and Gradient Consistency**
> > >
> > > We appreciate the reviewer’s attention to our reference to MultiDiffusion and its gradient consistency. To clarify:
> > >
> > > - MultiDiffusion empirically demonstrates the effectiveness of latent-space gradient manipulations in achieving coherent results in pixel space. For instance, diffusion path fusing, while operating in pixel space, leverages LDM’s latent-space gradients, indirectly demonstrating that latent-space gradient manipulations are impactful in producing coherent outputs.
> > > - While MultiDiffusion does not explicitly present a theoretical proof for gradient consistency, its experimental results imply that latent-space gradient manipulations contribute to pixel-space consistency. This empirical basis supports our use of similar gradient consistency principles in our method.
> > >
> > > We respectfully clarify that our paper’s core motivation is to address multiview semantic inconsistencies (e.g., the Janus problem) in text-to-3D generation by introducing **3D-consistent noising and gradient dissimilarity loss**. The cited works support the plausibility of latent-space gradient consistency in improving pixel-space outcomes, and our experiments (e.g., Figures 11 and 12) empirically demonstrate the relevance of this approach in LDM-based score distillation.
> > >
> > > We believe the limitations highlighted by “How I Warped Your Noise” regarding pixel-space fidelity do not undermine the semantic-level improvements we aim for. Instead, they reinforce the distinction between pixel-level fine details and high-level semantic consistency, which our method primarily targets.

---

### Official Review · Reviewer_cMr2 · 2024-10-31

**Soundness:** 2
**Presentation:** 2
**Contribution:** 2
**Rating:** 5
**Confidence:** 4

**Summary:**

This paper introduces a method called Geometry-aware Score Distillation. The experimental results demonstrate that the proposed approach can improve performance and address geometric inconsistencies in SDS-based text-to-3D generation tasks.

**Strengths:**

1. Efficient Incorporation of 3D Consistency: The paper directly adjusts the gradients of Score Distillation Sampling (SDS) without requiring 3D data to fine-tune a 2D diffusion model. This allows for the incorporation of 3D consistency with minimal computational overhead.

2. Significant Improvement over Baselines: The proposed method shows noticeable enhancements in both 3D consistency and appearance details when compared to baseline models such as GaussianDreamer, ProlificDreamer, and MVDream.

**Weaknesses:**

1. Dependency on Initial Point Cloud: The method requires establishing correspondences between different views, necessitating an initial point cloud. The paper primarily compares against Gaussian splatting-based methods like GaussianDreamer. According to previous literature [1], when shape initialization is used as a constraint, the Janus problem is substantially mitigated. Therefore, the effectiveness of the proposed method requires more rigorous justification.

2. Comparative Evaluation Concerns: In comparing their method with ProlificDreamer, the authors use 3D-Fuse—originally designed to address the Janus problem—as a baseline. This choice potentially diminishes the perceived effectiveness of the proposed method. Additionally, as observed in Figure 7, the method does not significantly enhance 3D consistency compared to ProlificDreamer; it primarily reduces floating artifacts in the background. Similar issues have been discussed in prior work [2] and can be addressed by adjusting the Classifier-Free Guidance (CFG) in diffusion models. Thus, further experimental evidence is needed to substantiate the authors' claim of enhanced 3D consistency.

[1] Chen, Cheng, et al. "Sculpt3d: Multi-view consistent text-to-3d generation with sparse 3d prior." Proceedings of the IEEE/CVF Conference on Computer Vision and Pattern Recognition. 2024.

[2] Yang, Xiaofeng, et al. "Learn to Optimize Denoising Scores: A Unified and Improved Diffusion Prior for 3D Generation." European Conference on Computer Vision. Springer, Cham, 2025.

**Questions:**

Please see the weakness part.

---

> ### Author Response · Authors · 2024-11-28
> **Response to Reviewer cMr2**
>
> **A1. Regarding dependency on Initial Point Cloud and 3DFuse.**
>
> We have already demonstrated in Figures 6 and 7 of our initially submitted paper on GaussianDreamer [1], ProlificDreamer [2], and Dreamfusion [3] baselines that **the Janus problem still remains even though they had incorporated Janus-reducing methods such as shape initializations or 3DFuse [4]**. We emphasize that our method effectively removes the geometric inconsistencies that remain even after applying these Janus-reducing methods.
>
> As noted by the reviewer, all of our baselines already either use initial point clouds or 3DFuse, which constrains SDS generation to the given point cloud shape to alleviate the Janus problem. However, our baseline results in Figures 6 and 13 of our revised paper clearly demonstrate the Janus problem (e.g. multiple faces in generated eagle in the result from the prompt "a DSLR photo of a majestic eagle", face in the back of monkey riding a cycle in the result from the prompt "zoomed out DSLR photo of a monkey riding a bike”), which shows there are cases where **using such initial shapes is not sufficient to remove the geometric inconsistency problem**.
>
> Our approach significantly alleviates these inconsistencies without any additional models or training, as shown in “our" results of Figures 6 and 13. This demonstrates the effectiveness of our method in addressing the limitations resulting from geometric inconsistencies, achieving higher-quality 3D generation. We will make this distinction more explicit in the revised manuscript and further highlight the Janus-removal effect of our method, shown in our qualitative results.
>
>  &nbsp;
>
> **A2. Our method functions without shape initialization or 3DFuse, and we demonstrate it with additional experiments.**
>
> Through additional experiments in **Figure 7 of our revised paper**, we also demonstrate that **our method performs effectively even without relying on initial point clouds or 3DFuse, highlighting that our method is not inherently dependent on them.** The experiments are conducted on ProlificDreamer baseline, and instead of using Point-E (to generate initial point cloud) or 3DFuse, we obtain our point cloud directly from volumetric-rendered depths of given viewpoints. This allows us to obtain point cloud aligned to 3D scene at every iteration without Point-E or initial point clouds, and we leverage this point cloud to conduct 3D consistent noising.
>
> The results demonstrate that our method remains effective in reducing the Janus problem improving the 3D geometry **even in scenarios where no Janus-reducing methods such as initial shapes or 3DFuse is used**. We observe consistent improvements in reducing geometric inconsistencies and addressing the Janus problem. As demonstrated, our approach effectively mitigates these inconsistencies solely through 3D-consistent noising and the gradient similarity loss, even in the absence of an explicit geometric prior.
>
>  &nbsp;
>
> [1] Yi, Taoran et al. “GaussianDreamer: Fast Generation from Text to 3D Gaussians by Bridging 2D and 3D Diffusion Models”, CVPR 2024
>
> [2] Wang, Zhenjyi et al. “ProlificDreamer: High-Fidelity and Diverse Text-to-3D Generation with Variational Score Distillation”, NeurIPS 2023
>
> [3] Poole, Ben et al. “DreamFusion: Text-to-3D using 2D Diffusion”, ICLR 2023
>
> [4] Seo, Junyoung et al. “Let 2D Diffusion Model Know 3D-Consistencyfor Robust Text-to-3D Generation”, ICLR 2024

---

### Official Review · Reviewer_F2bX · 2024-11-01

**Soundness:** 3
**Presentation:** 3
**Contribution:** 3
**Rating:** 6
**Confidence:** 3

**Summary:**

This paper focuses on optimization based text-to-3D generation. The authors propose a geometry-aware score distillation method to address the multi-view consistency issue in SDS.
Specifically, the authors propose to use 3D consistent noising by warping noises of nearby views with the current 3D modelling parameters. Based on this, a correspondence-aware gradient consistency loss is introduced to encourage the multiview consistency of SDS gradients between nearby viewpoints.

**Strengths:**

The paper addresses an important issue - multi-view inconsistency in SDS.
The method is well-motivated, and the design is intuitively sound.
The proposed method is shown to work well with various SDS variants, including 3DFuse, GaussianDreamer and ProlificDreamer.

The presentation is clear.
Good visualizations in Fig. 3, Fig. 4 and Fig. 5.

**Weaknesses:**

No video demos were shown to validate the generated 3D results.
Some of the generated results still appeal oversmoothness and unrealistic (Fig. 6, Fig. 11).

**Questions:**

Adding an algorithm highlighting the differences from the original SDS would help readers better understand the optimization process.

Since the paper claims to introduce minimal additional computational cost compared to SDS, more details regarding this cost should be included.

In addition to SDS-like optimization methods, there is another line of text-to-3D generation approaches that directly train on large-scale 3D datasets to enable feedforward generation, such as [1, 2]. Although these approaches are not directly comparable, clarifying this distinction in the related work section would be helpful.

Considering similar efforts to restrict the noise sampling space, a noise recalibration scheme was introduced in [3]. A discussion on the similarities and differences is recommended.

ref:

    [1] LGM: Large Multi-View Gaussian Model for High-Resolution 3D Content Creation.
    [2] 3DTopia: Large Text-to-3D Generation Model with Hybrid Diffusion Priors.
    [3] Diverse and Stable 2D Diffusion Guided Text to 3D Generation with Noise Recalibration.

---

> ### Author Response · Authors · 2024-11-28
> **Response to Reviewer F2bX**
>
> **W1. Adding an algorithm highlighting the differences from the original SDS would help readers better understand the optimization process.**
>
> **A1. Pseudocode algorithm included.**
>
> We have included a pseudocode in the Appendix B of the revised version of our paper. The pseudocode explicitly highlights the differences between the original SDS and our proposed method, especially detailing the incorporation of 3D-consistent noising in our SDS process. We believe this addition will significantly enhance the clarity of our optimization process and help readers better understand the key contributions of our approach.
>
>
>
> **W2. Question regarding computation cost.**
>
> **A2. Regarding computation cost.**
>
> Roughly, incorporating our methodology increases computation time by a factor of around 1.2 to the baseline. Specifically, our ProlificDreamer [1] baseline's optimization speed is average 1.198 it/sec (standard deviation=0.0116), while incorporation of our method slows it down to 1.016 it/sec (standard deviation=0.0045), yielding an training time increase by factor of 1.179. This additional computational cost is proportional to the training time of the original baseline and depends on the size of the point cloud being used for optimization. It is worth noting that the primary source of this increase is the inefficient rasterization code currently implemented, which we are actively improving through CUDA programming. Despite this marginal increase, our method demonstrates faster convergence, as evidenced in Figure 9 of our initially submitted paper, making the time required to achieve comparable quality in generated 3D scenes effectively similar. Furthermore, as a plug-and-play methodology, our approach offers substantial benefits with minimal computational overhead.
>
>
>
> **W3. Comparison to feedforward 3D generation methods required.**
>
> **A3. Additional comparison to feedforward 3D generation methods.**
>
> Our approach, leveraging SDS-like optimization, directly utilizes 2D diffusion priors to achieve 3D generation without requiring large-scale 3D datasets or expensive training specific to 3D. This capability enables us to benefit from the extensive and diverse knowledge encoded in 2D diffusion models, such as CLIP-guided latent space priors, enhancing the semantic and stylistic fidelity of the generated 3D models. By contrast, feedforward methods such as LGM [2] are constrained by the quality and diversity of the 3D datasets they are trained on, which can limit their generalization capabilities to novel or complex prompts. Leveraging these rich 2D priors, our method provides a flexible and efficient alternative, especially in scenarios where high-quality 3D data is scarce. Lastly, in our initially submitted paper, **we had already included a similar discussion in Appendix C, ”Analysis in Comparison to Multiview Generation Model”**, comparing our model to feedforward methods regarding the diversity and generation quality of generated results.
>
>
>
> **W4. Comparison to noise recalibration scheme introduced in [3].**
>
> **A4. Difference between our work and noise recalibration scheme [3].**
>
> The noising recalibration scheme in [3] can be summarized as making the noise used for denoising process in SDS learnable, thereby restricting the noise sampling process to a single
> random noise sampled from the Gaussian space. Our method largely differs from this method in that we do not make the noise learnable nor fixate it to a singular noise throughout optimization, but only aims to ensure the noise between different camera viewpoints remain consistent to one another. In this sense, the method described in [3] can be seen as orthogonal to ours, as fixing our 3D consistent noise and making it learnable would be effectively adding [3] to our method.
>
>
>
> [1] Wang, Zhenjyi et al. “ProlificDreamer: High-Fidelity and Diverse Text-to-3D Generation with Variational Score Distillation”, NeurIPS 2023
>
> [2] Tang, Jiaxiang et al. “LGM: Large Multi-View Gaussian Model for High-Resolution 3D Content Creation”, ECCV 2024
>
> [3] Yang, Xiaofeng et al. “Diverse and Stable 2D Diffusion Guided Text to 3D Generation with Noise Recalibration”, ArXiv 2024

---

### Meta-Review · Area_Chair_r6AW · 2024-12-21

**Metareview:**

Summary:
The paper aims to improve the geometry consistency of the the text-to-3D generation problem. Its technical contribution lies in a geometry-aware score distillation method, which includes geometry-based gradient warping and gradient dissimilarity loss.


Strength:
- Multiview consistency is a core problem in SDS.
- No model finetune or 3D data is required.
- Improvement over baseline models such as GaussianDreamer, ProlificDreamer, and MVDream.

Weakness:
- no video demo to show the 3D results
- lack of novelty for the 3D consistent noising
- the noising method is specific for point cloud and 3DGS methods; it is unclear how this can be applied to NeRF methods or mesh.
- Experimental results are weak, e.g., underperforming SoTA methods like ProlificDreamer and LucidDreamer.

Justification:

The paper received four reviews. Only one reviewer (Reviewer F2bX) is mildly positive, while the other three reviewers are mildly negative. After the rebuttal, all reviewers appreciated the efforts but remained unconvinced, particularly regarding the experimental evaluation. As these concerns have not been adequately addressed, the AC believes that it's not yet ready for publication.

**Additional Comments On Reviewer Discussion:**

Reviewer F2bX:

- During the rebuttal, the authors provided pseudocode in Appendix B to highlight the difference between the original SDS and the proposed method.

The authors provided computation costs, such as a training time increase of 1.179 with respect to ProlificDreamer.

The authors mentioned that the initial submission already included some discussions in Appendix C regarding comparisons with feedforward 3D generation.

Reviewer cMr2, Reviewer Wsmb, Reviewer C6Fp
- dependency on Initial Point Cloud and 3DFuse. Reviewer Wsmb also noted that shape initialization methods, such as 3D fuse, may substantially reduce the multiview inconsistency problem. However, the core claim that "consistent noising helps reduce the Janus problem" has not been fully validated. The AC thinks this is a crucial point.

Overall, while the reviewers appreciate the authors' response, their concerns regarding the experimental validation, the motivation behind 3D consistent noising, and lack of visuals remain. The AC agrees with the reviewers and finds no ground to accept at the current form.

---

### Decision · Program_Chairs · 2025-01-22

Reject